# On the Surprising Effectiveness of a Single Global Merging in Decentralized Learning

**Tongtian Zhu[1*], Tianyu Zhang[2,3*], Mingze Wang[4], Zhanpeng Zhou[5†], Can Wang[1]**

[1]Zhejiang University  [2]Mila, Quebec AI Institute  [3]Université de Montréal
[4]Peking University  [5]Shanghai Jiao Tong University

{raiden, wcang}@zju.edu.cn    tianyu.zhang@mila.quebec
mingzewang@stu.pku.edu.cn    zzp1012@sjtu.edu.cn

## Abstract

Decentralized learning provides a scalable alternative to parameter-server-based training, yet its performance is often hindered by limited peer-to-peer communication. In this paper, we study how communication should be scheduled over time, including determining when and how frequently devices synchronize. Counterintuitive empirical results show that concentrating communication budgets in the later stages of decentralized training substantially improves global test performance. Surprisingly, we find that fully connected communication at the final step, implemented as a single global merge, can significantly improve the performance of decentralized learning under high data heterogeneity. Our theoretical contributions, which explain these phenomena, are the first to establish that the globally merged model of decentralized SGD can match the convergence rate of parallel SGD. Technically, we reinterpret part of the discrepancy among local models, which was previously considered detrimental noise, as a constructive component essential for matching this rate. This work provides evidence that decentralized learning can generalize under high data heterogeneity and limited communication, while offering broad new avenues for model merging research. The blog post and code are available at 🔗 Grok in Decentralized Learning and 🐙 Code, respectively.

## 1 Introduction

Decentralized learning offers a promising approach to crowdsource computational workloads across geographically distributed computing resources (Yuan et al., 2022; Borzunov et al., 2023b; Jaghouar et al., 2024). A defining characteristic of this setting is the reliance on peer-to-peer communication during training, involving the peer-level exchange of model parameters or gradients. However, such communication is often constrained in practice due to limited bandwidth between geographically distant nodes, making it a scarce resource. These constraints can significantly degrade the performance of decentralized learning, both theoretically and empirically (Lian et al., 2017; Koloskova et al., 2020; Vogels et al., 2021). As a result, efficiently allocating limited communication resources becomes a fundamental challenge in decentralized learning, especially in heterogeneous environments where varying local data distributions intensify communication demands (Martínez Beltrán et al., 2023).

To date, most efforts addressing this challenge have focused on optimizing communication allocation at the *spatial level*, particularly through the design of communication graphs (Ying et al., 2021; Li et al., 2022b; Takezawa et al., 2023). In contrast, *temporal* communication allocation, i.e., deciding when and how frequently agents synchronize, remains a significant yet underexplored direction for improving decentralized learning. Although temporal communication allocation has been studied in federated learning (FL) (McMahan et al., 2017; Wang et al., 2019), this problem remains largely untouched in the fully decentralized setting, which is fundamentally different due to the lack of a central server for global aggregation (see Section 2 and Remark 1).

---

*Equal contribution. †Corresponding author.

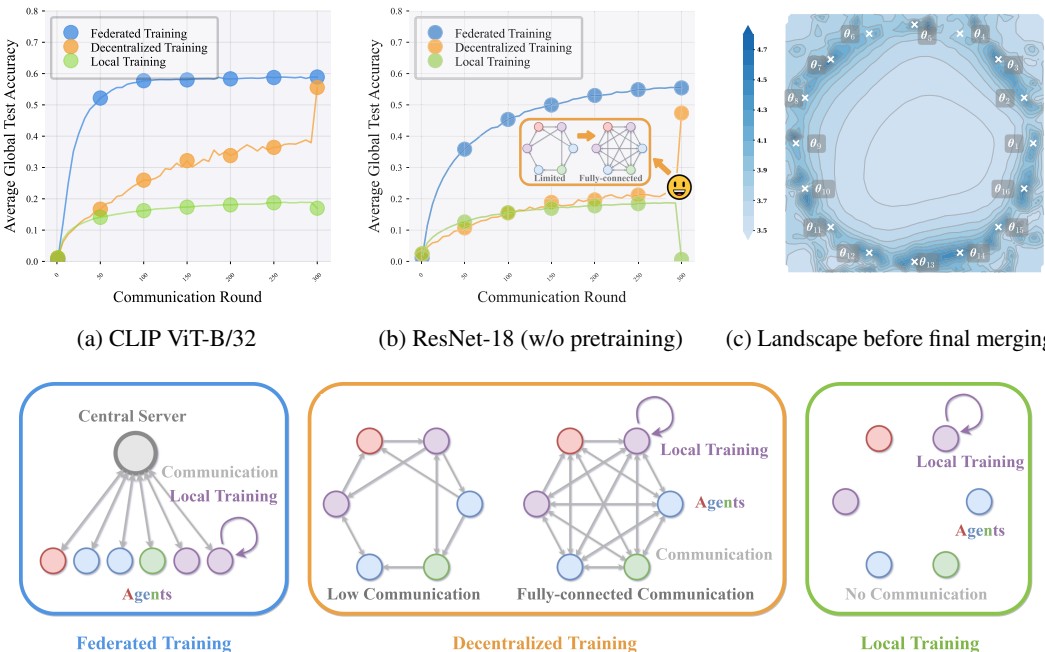

(a) CLIP ViT-B/32     (b) ResNet-18 (w/o pretraining)     (c) Landscape before final merging

(d) A comparative illustration of federated, decentralized, and local training.

Figure 1: **(a, b)** Global test accuracy (see Definition 1) of CLIP ViT-B/32 (a) and ResNet-18 (b) on Tiny ImageNet under a 32-agent non-IID setup (Dirichlet $\alpha = 0.1$), where decentralized SGD communicates with one random peer per round with probability 0.2 and performs a final global merge. **(c)** Loss landscape before the final merge for decentralized SGD.[3] **(d)** Illustration of federated, decentralized, and local training. Experimental details are provided in Appendix C.1.

> **Question**: *How should we allocate communication budgets over time in decentralized learning?*

To answer this question, we design a series of experiments that allocate communication budgets across different time windows during training (see Figure 2). Specifically, we divide the training process into consecutive windows, each with a fixed number of communication rounds. We assign higher communication budgets to selected windows using global synchronization via `AllReduce` (Sergeev & Del Balso, 2018), while otherwise keeping communication low through infrequent synchronization with random peers.[1] This design reveals how temporal communication allocation affects performance under constrained budgets. We observe that allocating higher communication budgets toward the later stages of training consistently leads to improved final test performance (see Definition 1). More surprisingly, we observe the remarkable effect of a single round of fully connected communication.[2]

> **Surprising Phenomenon**: *A single global merging of decentralized models, under severely constrained communication and high data heterogeneity, can significantly improve global test performance.*

**Our contributions** are summarized below.

- **Empirical Observations**. (**1**): We highlight the critical role of a single global merging in decentralized training, showing that it can achieve performance close to that of federated learning, even under severe communication constraints and data heterogeneity (see Figure 1a, Figure 1b). (**2**): We observe that limited but nonzero communication preserves a challenging cross-initialization, cross-distribution "mergeability" of local models throughout training (see Definition 2, Figure 1c,

---

[1]Agents refer to participants in decentralized learning. "Communication" and "synchronization" are used interchangeably.

[2]Fully connected communication refers to global synchronization via `AllReduce`. In this paper, fully connected communication is realized through parameter averaging over the models on all agents, namely *global merging*.

[3]We use 16 agents for loss landscape visualization to ensure visual clarity.

and the blue curve in Figure 2c), which does not hold under complete local training (green curve in Figure 1a, Figure 1b). These findings remain consistent across datasets, heterogeneity levels, model architectures, and communication topologies (see Appendix C.3), and provide a first systematic study of global merging in decentralized learning.

- **Theoretical Contributions**. We investigate the underlying mechanism that enables the *mergeability* of local models in decentralized learning. Specifically, we provide the first convergence analysis showing that the globally merged model of decentralized SGD can match the rate of parallel SGD (Theorem 1 and Proposition 2). Furthermore, we offer a theoretical explanation for why limited but nonzero communication can ensure mergeability, and why communication should be concentrated in the later stages of training (see Proposition 3).

We anticipate that this work will pave the way for principled decentralized training algorithms capable of generalizing under communication constraints and data heterogeneity, while also advancing model merging research (see the discussion in Section 6 and the **Q&A Section** in Appendix A).

## 2    RELATED WORK

**Temporal Communication Allocation in Parallel, Federated, and Decentralized Learning.**
Communication allocation is well-studied in both data-centric parallel learning (Li et al., 2014) and federated learning (FL) (McMahan et al., 2017). In parallel learning settings, Gu et al. (2024) proposed a novel strategy for scheduling local steps by analyzing the implicit bias of Local SGD (Gu et al., 2023). FL extends this server-based paradigm to handle not identically and independently distributed (non-IID) data, but it critically retains a global model. This reliance on a global model has shaped a broad consensus in the FL literature: frequent, early-stage communication is considered essential for aligning local models (Wang et al., 2019; Tang et al., 2020).

In contrast, our work addresses fully decentralized learning, a fundamentally different setting that lacks a central server. Instead of optimizing a generic global model, the goal is to make local models generalize to the global distribution. Despite extensive work focusing on communication allocation at the spatial level in decentralized learning (e.g., designing communication topologies) (Ying et al., 2021; Li et al., 2022b; Takezawa et al., 2023; Kharrat et al., 2024), few studies examine temporal allocation; a pioneering IID study (Kong et al., 2021) showed that stronger early alignment to the global average can modestly improve test performance. These findings do not directly transfer to non-IID settings, where $\mathcal{L}(\cdot) \equiv \mathcal{L}_k(\cdot)$ no longer holds (see Equation (1) and Definition C.2), so they mainly characterize local rather than global generalization (see Definition 1). Due to space constraints, we refer readers to Appendix B.2 and Appendix B.3 for related work on the implicit bias of decentralized learning, and on the topic of model merging.

## 3    NOTATIONS AND PRELIMINARIES

### 3.1    NON-IID DECENTRALIZED LEARNING

Decentralized learning formalizes distributed learning as an optimization problem over a connected graph $G = (\mathcal{V}, \mathcal{E})$, where $\mathcal{V}$ contains $m$ agents and $\mathcal{E}$ denotes the communication links. Each agent $k \in \mathcal{V}$ samples data from a local distribution $\mathcal{D}_k$ and maintains a local model $\theta_k \in \mathbb{R}^d$. The objective is to learn a consensus model $\theta$ that minimizes the global population risk (Koloskova et al., 2020):

$$\min_{\theta \in \mathbb{R}^d} \left[ \mathcal{L}(\theta) \triangleq \frac{1}{m} \sum_{k \in \mathcal{V}} \mathbb{E}_{\xi_k \sim \mathcal{D}_k} \mathcal{L}(\theta; \xi_k) \right], \tag{1}$$

where $\mathbb{E}_{\xi_k \sim \mathcal{D}_k} \mathcal{L}(\theta; \xi_k) \triangleq \mathcal{L}_k(\theta)$ denotes the local population risk of $\theta$ for an unseen sample $\xi_k \sim \mathcal{D}_k$.

In practice, the optimization of Equation (1) is performed under the empirical risk minimization framework, leveraging $m$ local datasets $S \triangleq \bigcup_{k=1}^{m} S_k$, where $S_k = \{\xi_{k,1}, \dots, \xi_{k,\zeta}\}$ denotes the dataset of agent $k$ sampled from $\mathcal{D}_k$. The resulting optimization problem is given by:

$$\min_{\theta \in \mathbb{R}^d} \left[ \mathcal{L}_S(\theta) \triangleq \frac{1}{m} \sum_{k \in \mathcal{V}} \sum_{\zeta=1}^{n_k} \mathcal{L}(\theta; \xi_{k,\zeta}) \right]. \tag{2}$$

To solve the optimization problem in Equation (2), decentralized algorithms minimize the global empirical risk with only local computation and peer-to-peer communication (Tsitsiklis et al., 1986; Nedic & Ozdaglar, 2009). The communication graph is governed by a weighted adjacency matrix $W^{(t)} \in [0, 1]^{m \times m}$, sampled from a distribution $\mathcal{W}^{(t)}$, where each entry $W_{k,l}^{(t)} \geq 0$ reflects the influence of agent $l$ on agent $k$.[4] Decentralized learning algorithms operate by alternating between local updates and model aggregation through communication with neighbors, as outlined in Algorithm 1.

---

**Algorithm 1** Decentralized Learning

---

**input** Initialize values $\theta_k^{(0)} \in \mathbb{R}^d$ on each agent $k \in \mathcal{V}$, number of steps $T$, mixing matrix $W$

1: **in parallel on all agents $k \in \mathcal{V}$, for $t = 0, \ldots, T - 1$ do**
2:    Sample training data $\xi_k^{(t)}$ from $\mathcal{D}_k$, $\theta_k^{(t+\frac{1}{2})} \leftarrow Optimizer(\theta_k^{(t)}, \xi_k^{(t)})$           ▷ Local update
3:    Send $\theta_k^{(t)}$ to out-neighbor(s) and receive $\{\theta_l^{(t)}\}_{l \in \mathcal{N}_{\text{in}}(k)}$ from in-neighbor(s) ▷ Communication
4:    Sample mixing matrix $W^{(t)} \sim \mathcal{W}^{(t)}$, $\theta_k^{(t+1)} \leftarrow \sum_{l \in \mathcal{N}_{\text{in}}(k)} W_{k,l}^{(t)} \theta_l^{(t+\frac{1}{2})}$     ▷ Gossip averaging
5: **end parallel for**

---

**Practical Evaluation Metrics**. In decentralized learning, data heterogeneity and limited training time often prevent a full consensus model $\theta$. We therefore propose to adopt *average global test accuracy*, a proxy for global population risk, as the primary metric to quantify how well local models generalize to the global data distribution.

**Definition 1** (Average Global Test Accuracy). *The average accuracy of agents $k \in \mathcal{V}$ is defined as:*

$$\underbrace{\overline{\text{Acc}}(\{\theta_k^{(t)}\}_{k \in \mathcal{V}}) = \frac{1}{m} \sum_{k \in \mathcal{V}} \text{Acc}(\theta_k^{(t)}),}_{\textit{Average Accuracy across agents}} \quad \textit{where} \ \ \text{Acc}(\cdot) \triangleq \underbrace{\frac{1}{m} \sum_{l \in \mathcal{V}} \mathbb{E}_{\xi_l \sim \mathcal{D}_l} \text{Acc}(\cdot; \xi_l)}_{\textit{Test accuracy on the global distribution}} \ .$$

> **Remark 1** (Metric Justification). This metric is specifically designed to address a core question in fully decentralized learning: *how well do local models $\{\theta_k^{(t)}\}_{k \in \mathcal{V}}$, trained with limited peer-to-peer synchronization, generalize to the global distribution $\mathcal{D}$?* This metric offers a more realistic evaluation for decentralized settings without a global model. See the discussion in Appendix C.2.

### 3.2 MERGEABILITY

**Definition 2** (Mergeability under Global Population Risk). *A set of local models $\{\theta_k\}_{k \in \mathcal{V}}$ is globally mergeable if there exist combination weights $\{w_k\}_{k \in \mathcal{V}} \in [0, 1]$ such that:*

$$\mathcal{L}\left(\sum_{k \in \mathcal{V}} w_k \theta_k\right) \leq \sum_{k \in \mathcal{V}} w_k \mathcal{L}(\theta_k), \tag{3}$$

*where $\mathcal{L}(\cdot)$ denotes the global population risk.*

Definition 2 formalizes the intuition that a linearly interpolated model performs no worse than the original local models. This definition is inherently non-trivial due to the *non-convexity* of $\mathcal{L}$.

## 4 EMPIRICAL OBSERVATIONS

### 4.1 INCREASING IMPACT OF COMMUNICATION IN THE LATER STAGES OF TRAINING

To investigate potential solutions to communication scheduling, we explore a direct strategy: concentrating communication in a small subset of communication rounds. To this end, we divide the training process into consecutive windows, each consisting of a fixed length of communication rounds.

---

[4]Our framework incorporates a randomized decentralized learning setting where the weighted adjacency matrix $W^{(t)}$ can change during training (Boyd et al., 2006; Koloskova et al., 2020; Vos et al., 2023).

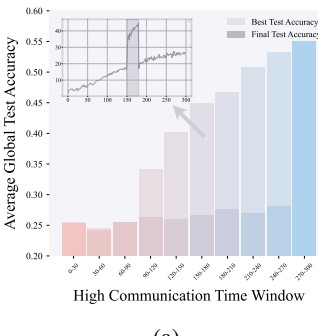 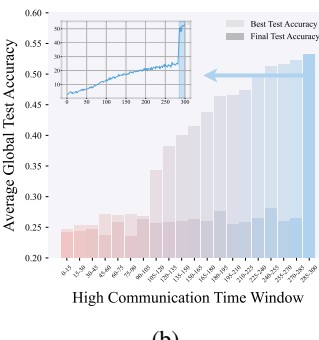 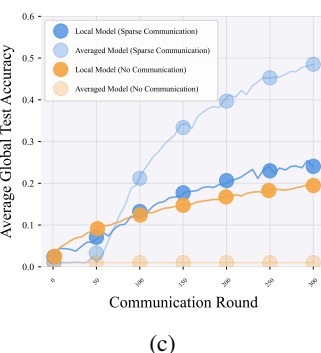

|     |     |     |
| :-: | :-: | :-: |
| (a) | (b) | (c) |

Figure 2: **(a, b)**: Comparisons of global test accuracy (see Definition 1) in decentralized training of ResNet-18 on CIFAR-100 with AdamW, distributed across 16 agents with Dirichlet $\alpha = 0.1$ (see Appendix C.1). Fully connected communication (i.e., AllReduce) is activated only in specific windows, while low communication, in which each agent communicates with one random peer with probability 0.2, is used elsewhere. **(a)**: Fully connected communication in $1/10$ of total rounds. **(b)**: Fully connected communication in $1/20$ of total rounds. In both, lighter bars show peak accuracy, darker bars show final accuracy. **(c)**: Global test accuracy for local models and the globally averaged model (counterfactual) under persistent low communication (blue) and no communication (orange).[7]

Specifically, the communication scheme is as follows: (1) fully connected communication (see Figure 1d (b)) is activated only within specific communication windows (i.e., global synchronization via `AllReduce` (Sergeev & Del Balso, 2018)[5]); (2) in all other rounds, each agent communicates only with *one* random peer with probability 0.2 (see "Communication Graph" in Appendix C.1).[6]

As shown in Figure 2, training is divided into 10 and 20 communication windows in (a) and (b), respectively. The bars in Figure 2 show both the best global test accuracy achieved during training (lighter-colored bars) and the final test accuracy at the end of training (darker-colored bars). Each bar corresponds to one communication window, where fully connected communication is applied *only* to the rounds within that window, while random peer communication is used in all other rounds. For instance, the inset in Figure 2a presents the complete test accuracy trajectory when fully connected communication is applied during rounds 150 to 180. A consistent trend emerges: *allocating communication budgets toward the later stages of training yields substantial improvements, particularly in final test accuracy.*

## 4.2 A SINGLE GLOBAL MERGING SIGNIFICANTLY IMPROVES GLOBAL TEST PERFORMANCE

In Figure 2b, we reduce the fully connected communication window length to 10 rounds, yet still observe substantial improvements in global test performance. This observation naturally raises the question: *What happens if the fully connected window is reduced to a single round?*

To investigate this, we conduct experiments where fully connected communication is applied only once, implemented by a single global merging. As shown in Figure 1a and Figure 1b, a single global merging is sufficient to significantly improve global test performance. Consistent gains are observed across a wide range of settings, including different datasets, architectures, and communication topologies (see additional experiments in Appendix C.3). The significant increase in performance suggests that the potential of decentralized learning might be considerably underestimated.

**Comparisons.** D-PSGD (Lian et al., 2017) introduced final global merging under IID settings, but did not analyze the performance gap before and after merging. In contrast, we provide a systematic study of this recovery in challenging non-IID scenarios. Chen et al. (2021) showed the benefit of periodic global averaging, but their method requires frequent global communication (every $H = 48$ steps), whereas we recover performance with only *a single* merging. We also compare with Skew-

---

[5]We note that `AllReduce` can be efficiently realized in a decentralized manner such as `Ring-ALLReduce`.

[6]**Code**: https://github.com/Raiden-Zhu/ICLR-2026-Grokking-in-Decentralized-Learning

[7]The term "counterfactual" refers to the fact that no global merging occurs during decentralized training. Instead, we manually compute the test accuracy of the hypothetical globally averaged model to quantify the "mergeability" of local models.

Compensated Sparse Push (SCSP) (Aketi et al., 2021), which also includes a final global merging step. While both works aim to reduce communication, they differ from ours in methodology and setting: (1) SCSP uses *gradient sparsification* (top-$k$ gradients) over a fixed topology, whereas we study mergeability under *topological sparsification* (sparse gossip); (2) SCSP focuses on one local step ($H = 1$), whereas we show robust mergeability with multiple local steps (e.g., $H = 100$) under high heterogeneity. While these works share the broader goal of improving communication efficiency, our work offers a new perspective by investigating the *mergeability* itself: why local models retain this property despite extremely limited communication and high data heterogeneity.

---

**Remark 2** (Non-Triviality of the Performance Gain). One may attribute the final improvement to the cumulative effect of sparse gossip. In our setup, each communication round activates one random peer exchange with probability $R = 0.2$ over $T = 300$ rounds, yielding about 60 peer exchanges per agent in expectation. This may appear analogous, on average, to multiple implicit global aggregations. If this additive interpretation were sufficient to explain the gain, local models should already be better aligned and achieve performance close to that of the post-merge model. However, our empirical results challenge this interpretation: as shown in Figure 1b, local models under sparse gossip still exhibit poor global test performance, close to the no-communication case, while a single final global merging produces a large performance gain. This gap indicates that the effect of single merging is non-trivial, rather than a simple result of accumulated local gossip.

---

**Communication Cost Comparison.** Let $P$ be the model size, $m$ the number of agents, and $T$ the number of training rounds. A standard `Ring-AllReduce`-based protocol incurs a total communication cost of $\mathcal{O}(2mPT)$. In contrast, our decentralized setup incurs a cost of $\mathcal{O}(mRPT + 2mP)$, where $R \ll 2$ denotes the expected number of peers per round, and the $\mathcal{O}(2mP)$ term arises from final global merging via `Ring-AllReduce`. We also note that while a global merging may appear less practical in some cases, it can be effectively approximated via multiple rounds of gossip synchronization among local agents (see our supplementary experiments in Appendix C.3.4).

## 4.3 MERGEABILITY PERSISTS UNDER LIMITED BUT NONZERO COMMUNICATION

A follow-up question is whether the effectiveness of global merging is specific to the end of training. To investigate this, we assess the *counterfactual* performance of the globally averaged model at each training round, as depicted by the light-blue curve in Figure 2c. The experiments are conducted under a lower-communication setting, where each agent communicates with one random peer at each round with probability 0.2 (see "Communication Graph" in Subsection C.1). A consistent superiority of the merged model (light-blue curve) over the local models (dark-blue curve) is observed throughout training, suggesting that local models remain mergeable at all stages (see Definition 2).

As an ablation, we conduct an experiment where all models are trained fully locally, without communication (see Figure 2c). In this case, the counterfactual test performance of the globally averaged model remains close to zero (light-orange curve), indicating that without communication, local models are not mergeable. This suggests that mergeability does *not* arise inherently from the local models themselves. Interestingly, under the low-communication setting, the performance of local models before merging (dark-blue curve) remains similar to that in the no-communication case (dark-orange curve). However, after global merging, the models show significant performance improvement. This contrast clearly implies that extremely limited but nonzero communication enables mergeability.

**Mergeability without Consensus.** Prior work on gossip algorithms has suggested that local models may converge to a similar state even under minimal communication (Jelasity et al., 2005). In contrast, our work addresses a more challenging heterogeneous data setting, in which local models do not reach a single consensus point yet remain mergeable. Specifically, we identify an emergent geometric structure where decentralized training guides local models to a ring-like high-loss region surrounding a central low-loss basin (see Figure 1c). Notably, this corresponds to a more challenging cross-initialization, cross-distribution merging scenario, where local models do not start from a shared pretrained checkpoint and are trained on non-IID data distributions.

## 5 THEORETICAL ANALYSIS

In this section, we examine the underlying mechanisms that enable the mergeability of local models in decentralized learning. As an initial step, we conduct a fine-grained convergence analysis of the globally merged models trained by Decentralized SGD (DSGD).[8] To substantiate the mergeability of local models, we compare the convergence rate of the merged model trained by DSGD with that of parallel SGD. Remarkably, we prove that the merged model in decentralized learning can match the rate of parallel SGD (Dekel et al., 2012; Li et al., 2014). This supports the empirical findings that the merged model can preserve the performance of individual local models (see Definition 2).

### 5.1 ASSUMPTIONS

We start by introducing the commonly used assumptions (Kong et al., 2021; Koloskova et al., 2020).

**Assumption 1** (Mixing matrix). *Each sample of the (randomized) mixing matrix $W \in \mathbb{R}^{m \times m}$ is doubly stochastic. Moreover, there exists $p > 0$ such that*

$$\mathbb{E}_W \|\Theta W - \bar{\Theta}\|_F^2 \leq (1-p)\|\Theta - \bar{\Theta}\|_F^2, \ \forall \Theta \in \mathbb{R}^{d \times m}. \tag{4}$$

*Here $\Theta = [\theta_1, \ldots, \theta_m]$, $\bar{\Theta} = [\bar{\theta}, \ldots, \bar{\theta}] \equiv \Theta \frac{1}{m} \mathbf{1}\mathbf{1}^\top$ where $\bar{\theta} = \frac{1}{m}\sum_{k=1}^m \theta_k$.*

**Assumption 2** (Smoothness). *The objective function $\mathcal{L}$ is four times continuously differentiable (i.e., $\mathcal{L} \in \mathcal{C}^4$) and there exist constants $L_q \geq 0$ for $q \in \{1, \ldots, 4\}$ such that:*

$$\|\nabla^q \mathcal{L}(\theta)\| \leq L_q, \quad \forall \theta \in \mathbb{R}^d. \tag{5}$$

We note that given $\mathcal{L} \in \mathcal{C}^2$, the boundedness of the Hessian norm (i.e., the case $q = 2$) implies that $\mathcal{L}$ is $L_2$-smooth, thereby recovering Assumption D.1 with $L = L_2$.

**Assumption 3** (Bounded noise and diversity). *There exist $\sigma^2, \zeta^2 \geq 0$ such that for any $\theta_k \in \{\theta_k\}_{k=1}^m$:*

$$\frac{1}{m}\sum_{k=1}^m \mathbb{E}_{\xi_k} \|\nabla \mathcal{L}_k(\theta_k; \xi_k) - \nabla \mathcal{L}_k(\theta_k)\|_2^2 \leq \sigma^2, \quad \frac{1}{m}\sum_{k=1}^m \|\nabla \mathcal{L}_k(\theta_k) - \nabla \mathcal{L}(\theta_k)\|_2^2 \leq \zeta^2, \tag{6}$$

*where $\mathcal{L}(\theta) = \frac{1}{m}\sum_{k=1}^m \mathcal{L}_k(\theta)$.*

Here $\sigma$ measures the local noise level and $\zeta$ measures the heterogeneity among agents.

### 5.2 CONVERGENCE ANALYSIS

**Theorem 1** (Non-convex Convergence Rate of DSGD). *Suppose Assumption 2 and Assumption 3 hold. Consider decentralized SGD (DSGD) with initializations $\theta_k^{(0)} = \theta^{(0)}$ for all $k \in \mathcal{V}$, and a constant learning rate satisfying $\eta < \frac{2}{L_2}$. Let $\bar{\theta}^{(t)} = \frac{1}{m}\sum_{k=1}^m \theta_k^{(t)}$ denote the averaged model at the $t$-th step. To achieve an $\varepsilon$-stationary point such that $\frac{1}{T}\sum_{t=0}^{T-1} \mathbb{E}\big[\|\nabla \mathcal{L}(\bar{\theta}^{(t)})\|_2^2\big] \leq \varepsilon$, the total number of steps $T$ satisfies:*

$$T = \mathcal{O}\Big(\tfrac{\sigma^2}{m\varepsilon^2} + \tfrac{1}{\varepsilon} + \sum_{t=0}^{T-1} U^{(t)}\Big) \cdot \big(\mathcal{L}(\theta^{(0)}) - \mathcal{L}^\star\big),$$

*where $U^{(t)} = (\eta L_2 - 1)(\nabla \mathcal{L}(\bar{\theta}^{(t)}))^\top \nabla \operatorname{Tr}\big(\nabla^2 \mathcal{L}(\bar{\theta}^{(t)}) \Gamma^{(t)}\big) + \Theta(\Xi_t^3)$, with $\Gamma^{(t)} = \frac{1}{m}\sum_{k=1}^m (\theta_k^{(t)} - \bar{\theta}^{(t)})(\theta_k^{(t)} - \bar{\theta}^{(t)})^\top$ and the consensus distance $\Xi_t^2 = \operatorname{Tr}(\Gamma^{(t)})$.*

**Remark 3.** We note that Theorem 1 gives an implicit bound that depends on $U^{(t)}, t \in \{1, 2, \ldots, T-1\}$, rather than a closed-form expression. It primarily serves to bridge convergence with the per-iteration dynamics of $U^{(t)}$, facilitating the subsequent derivation of the conditions on consensus and communication required to recover the parallel SGD rate (see Proposition 2 and Proposition 3).

---

[8]DSGD refers to standard decentralized SGD where the optimizer in Algorithm 1 is replaced with SGD.

Table 1: Comparison of non-convex convergence rates for parallel SGD and DSGD, both run with $m$ agents under non-IID data.

| Algorithm | Parallel SGD | DSGD (Koloskova et al., 2020) | DSGD (ours) |
|---|---|---|---|
| Rate | $\mathcal{O}\left( \frac{\sigma^2}{m\varepsilon^2} + \frac{1}{\varepsilon} \right)$ | $\mathcal{O}\left( \frac{\sigma^2}{m\varepsilon^2} + \frac{1}{p\varepsilon} + \frac{\sqrt{p}\,\sigma + \zeta}{p\,\varepsilon^{3/2}} \right)$ | $\mathcal{O}\left( \frac{\sigma^2}{m\varepsilon^2} + \frac{1}{\varepsilon} + \frac{1}{\varepsilon}\sum_{t=0}^{T-1} U^{(t)} \right)$ |

**Comparison and Technical Novelty**. As summarized in Table 1, the unified analysis of Koloskova et al. (2020) showed that DSGD suffers from additional terms of order $\mathcal{O}\left( \frac{1-p}{p\varepsilon} + \frac{\sqrt{p}\,\sigma + \zeta}{p\,\varepsilon^{3/2}} \right)$ in the convergence rate compared to parallel SGD. The core idea behind their analysis is to separate the effects of three key factors: the descent force (i.e., the squared gradient norm), gradient noise, and parameter discrepancy among agents. Each of these components is then analyzed and controlled separately. Among them, both the gradient noise and the model discrepancy are treated as detrimental to convergence. In contrast, we adopt a new proof framework that leverages the implicit bias of decentralized learning (see Proposition D.3 (Zhu et al., 2023b) and Appendix B.2). Rather than treating the discrepancy among agents purely as noise, we partially incorporate it as a constructive component essential for matching the rate of parallel SGD. This intuition is formalized through the convergence guarantee provided in Theorem 1, which introduces an additional term of $\mathcal{O}\left( \frac{1}{\varepsilon}\sum_{t=0}^{T-1} U^{(t)} \right)$. In what follows, we conduct a fine-grained analysis of the sign of $U^{(t)}$.

**Remark 4** (Reduction to Standard Rates). We consider two special cases where the term $U^{(t)}$ vanishes because the consensus error is identically zero ($\Xi_t \equiv 0$):

- The single-agent case ($m = 1$);

- The synchronous Parallel SGD case, where perfect synchronization ensures $\theta_k^{(t)} \equiv \bar{\theta}^{(t)}$ for all $k$.

In both settings, the auxiliary term $U^{(t)}$ in Theorem 1 is exactly zero. Consequently, Theorem 1 naturally recovers the rate of standard (Parallel) SGD, which is of the order $\mathcal{O}\left( \frac{\sigma^2}{m\varepsilon^2} + \frac{1}{\varepsilon} \right)$.

To better characterize how the higher-order loss landscape affects the dynamics of $U^{(t)}$, we introduce a new assumption that is theoretically novel yet empirically supported by prior literature.

**Assumption 4** (Progressive sharpening). *For any positive semi-definite matrix $\Sigma$, the gradient of the population risk negatively aligns with the gradient of sharpness. Formally, $\forall \theta \in \mathbb{R}^d$,*

$$\nabla\mathcal{L}(\theta)^\top \nabla \operatorname{Tr}(\nabla^2\mathcal{L}(\theta)\Sigma) < 0. \tag{7}$$

By the density of $\mathbb{R}$, there exists a $\gamma > 0$ such that $\nabla\mathcal{L}(\theta)^\top \nabla \operatorname{Tr}(\nabla^2\mathcal{L}(\theta)\Sigma) < -\gamma\|\nabla\mathcal{L}(\theta)\| < 0$ holds strictly. We refer to $\gamma^* \triangleq \sup\left\{\gamma > 0 \mid \nabla\mathcal{L}(\theta)^\top \nabla\operatorname{Tr}(\nabla^2\mathcal{L}(\theta)\Sigma) < -\gamma\|\nabla\mathcal{L}(\theta)\|\right\}$ as the *degree of progressive sharpening*.

**Remark 5**. $\operatorname{Tr}(\nabla^2\mathcal{L}(\theta)\Sigma)$ can be interpreted as an "average sharpness" around $\theta$; see similar metrics in (Gu et al., 2023; Zhu et al., 2023b). Assumption 4 reflects a widely observed phenomenon in deep learning: the loss gradient exhibits a negative correlation with the gradient of sharpness (Wang et al., 2022; Damian et al., 2023; Cohen et al., 2025). Intuitively, this condition implies that as the optimizer moves to reduce the loss, it simultaneously moves in a direction that increases the sharpness.

Assumption 4 ensures $\nabla\mathcal{L}(\bar{\theta}^{(t)})^\top \nabla \operatorname{Tr}\left(\nabla^2\mathcal{L}(\bar{\theta}^{(t)})\,\Gamma^{(t)}\right)$ in $U^{(t)}$ remains negative. In the following, we establish that this term can dominate the other terms in $U^{(t)}$, thereby ensuring $U^{(t)} < 0$.

**Proposition 2.** *Suppose Assumption 2 and Assumption 4 hold. Assume $\eta > 1/L_2$, and assume $\|\nabla\mathcal{L}(\bar{\theta}^{(t)})\| \geq \mu_t > 0$ for all $t$. Consider the matrix $\Gamma^{(t)} = \frac{1}{m}\sum_{k=1}^m (\theta_k^{(t)} - \bar{\theta}^{(t)})(\theta_k^{(t)} - \bar{\theta}^{(t)})^\top$ and its trace $\Xi_t^2 = \operatorname{Tr}(\Gamma^{(t)})$. Then, for any fixed $m > 0$, there exists $\Xi_t^2 > 0$ such that*

$$U^{(t)} \triangleq \frac{1}{2}(\eta L_2 - 1)\nabla\mathcal{L}(\bar{\theta}^{(t)})^\top \nabla \operatorname{Tr}\left(\nabla^2\mathcal{L}(\bar{\theta}^{(t)})\,\Gamma^{(t)}\right) + O(\Xi_t^3) < 0. \tag{8}$$

**Explanation of Assumptions.** We assume a lower bound on the global gradient norm at the averaged parameters $\bar{\theta}^{(t)}$, i.e., $\|\nabla \mathcal{L}(\bar{\theta}^{(t)})\| \geq \mu_t > 0$. This applies to the global-data gradient, which can remain significant even if individual local gradients vanish. The assumption is motivated by the Polyak-Lojasiewicz (PL) condition (Polyak, 1963; Karimi et al., 2016), $\frac{1}{2}\|\nabla\mathcal{L}(\theta)\|^2 \geq \mu(\mathcal{L}(\theta) - \mathcal{L}^*)$, which bounds the gradient away from zero before reaching the optimum. Our assumption formalizes this pre-convergence lower bound at iteration $t$ as $\frac{1}{2}\mu_t^2$. We also note that Assumption 2 bounds loss derivatives up to fourth order, $\|\nabla^q\mathcal{L}(\theta)\| \leq L_q$ for $q = 1, 2, 3, 4$, which is necessary to analyze the interaction between the consensus error $\Xi_t$ and higher-order landscape.

At a high level, Proposition 2 highlights the potential of leveraging decentralized training to accelerate distributed training beyond communication efficiency. This theoretical insight aligns with the empirical gains observed in Figure C.1a. Specifically, Proposition 2 implies that satisfying $U^{(t)} < 0$ necessitates careful control of both $\eta$ and $\Xi_t^2$. We first discuss the learning-rate constraint and then analyze the role of $\Xi_t^2$, based on which we provide a theoretical justification for allocating more communication in the later stages of training in Subsection 5.3.

> **Remark 6** (Potential Acceleration Under Larger Learning Rate). Note that the coefficient of the term $\nabla\mathcal{L}(\bar{\theta}^{(t)})^\top \nabla \mathrm{Tr}(\nabla^2\mathcal{L}(\bar{\theta}^{(t)})\Gamma^{(t)})$ in Inequality (8) is $\frac{1}{2}(\eta L_2 - 1)$. To ensure the negativity of this term, we require the condition $\eta > \frac{1}{L_2}$. Notably, the resulting interval $\frac{1}{L_2} < \eta < \frac{2}{L_2}$ coincides exactly with the regime of "oscillatory convergence" in classical optimization theory for a quadratic objective $\mathcal{L}(\theta) = \frac{1}{2}\theta^\top H\theta$ (Polyak, 1987, Chapter 1, p. 26).

To provide intuition, we outline the proof sketch of Theorem 1 below. This analysis establishes a novel descent lemma tailored for decentralized SGD, demonstrating how learning rate control harnesses progressive sharpening to offset the negative effects of consensus error.

$$
\mathbb{E}_{\xi^{(t)}}\left[\mathcal{L}(\bar{\theta}^{(t+1)})\right] \leq \mathcal{L}(\bar{\theta}^{(t)}) - \underbrace{\left(\eta - \frac{\eta^2 L_2}{2}\right)}_{>0} \underbrace{\left\|\nabla\mathcal{L}(\bar{\theta}^{(t)})\right\|^2}_{\text{Standard Descent}}
$$
$$
+ \underbrace{\left(\eta^2 L_2 - \eta\right)}_{>0} \underbrace{\nabla\mathcal{L}(\bar{\theta}^{(t)})^\top \nabla\mathrm{Tr}(\nabla^2\mathcal{L}(\bar{\theta}^{(t)})\Gamma^{(t)})}_{<0, \text{ progressive sharpening}} + \frac{\eta^2 L_2 \sigma^2}{2m} + \mathcal{O}(\Xi_t^3). \quad (9)
$$

We note that this third-order effect differs from prior analyses, which were typically established in a near-minimum regime (Li et al., 2022c). By contrast, we show that this mechanism emerges whenever local models are inconsistent, i.e., $\Xi_t > 0$.

Equation (9) further illuminates the critical role of the consensus violation term $\Xi_t = \sqrt{\mathrm{Tr}(\Gamma^{(t)})}$. Crucially, the progressive sharpening term scales with $\mathcal{O}(\Xi_t^2)$, whereas the higher-order residual error is $\mathcal{O}(\Xi_t^3)$. Consequently, provided $\Xi_t$ is properly controlled such that the $\mathcal{O}(\Xi_t^2)$ gain dominates the $\mathcal{O}(\Xi_t^3)$ error, decentralized SGD can match or even surpass the convergence rate of parallel SGD. According to Corollary D.2, $\mathbb{E}[\Xi_t^2]$ is bounded by

$$
\mathbb{E}[\Xi_t^2] \leq \mathcal{O}\left(\frac{(1-p)}{p^2}\right), \quad (10)
$$

where the parameter $p$ (with $p \in (0, 1]$) reflects connectivity in the communication graph (see Assumption 1). A larger $p$ indicates better connectivity and faster consensus, while a smaller $p$ implies a sparse communication graph (i.e., lower communication) and slower information propagation. For example, $p = 1$ corresponds to a fully connected topology, enabling perfect communication, whereas $p = 0$ represents the extreme case of complete local training with no communication.

**Why Limited but Nonzero Communication Enables Mergeability.** Notably, random communication graphs can achieve $p = \Theta(1)$, striking a favorable trade-off: they require relatively low communication overhead while still maintaining efficient information mixing due to randomized edge sampling, which ensures a rapid decrease of $\Xi_t$ (Vos et al., 2023). This is why we adopt random topologies as the primary setup in our experiments: they can satisfy the condition in Proposition 2 even under extremely limited communication, thereby ensuring mergeability (see Figure 1). However, in the case of full local training where $p = 0$ (see Figure 1d), the right-hand side of Equation (10)

increases to infinity, indicating that $\Xi_t$ may diverge. As a consequence, the condition on $\Xi_t$ in Proposition 2 can no longer be satisfied, which explains why local models after complete local training cannot be reliably merged (see the green curve in Figure 1b).

## 5.3 A THEORETICAL EXPLANATION FOR COMMUNICATION ALLOCATION

Proposition 2 highlights the importance of $\Xi_t^2$ in satisfying Inequality (8). This motivates the question of how small $\Xi_t^2$ (or how large $p$) should be, which we answer in the following sufficient condition.

> **Proposition 3** (Critical Consensus Edge). *Suppose Assumption 1-Assumption 4 hold. Assume $\eta > \frac{1}{L_2}$ and that the consensus error satisfies $\Xi_t \leq 1$ for all $t$. Then the following condition ensures that the critical condition in Inequality (8) is satisfied:*
>
> $$\sqrt{\frac{24\,(1-p)\,\eta^2}{p^2}\left(\phi^2 + \sigma^2\right)} < \min\left\{\frac{(\eta L_2 - 1)\gamma^*\,\mu_t}{2(\eta L_2 + \frac{L_4}{24})\sqrt{m}L_1},\quad \sqrt{\frac{(\eta L_2 - 1)\gamma^*\,\mu_t}{2\Sigma_{\text{high}}}}\right\}, \quad (11)$$
>
> *where $\Sigma_{\text{high}} = \frac{1}{8}\eta L_2 L_3^2 + \frac{1}{2}\eta\sqrt{m}L_2 L_3 + \frac{\eta m L_2 L_4^2}{1152}$. Here, $\gamma^*$ denotes the degree of progressive sharpening (see Assumption 4), $\phi^2$ denotes the uniform upper bound of the averaged squared local gradient norm (i.e., $\frac{1}{m}\sum_{k=1}^{m}\|\nabla\mathcal{L}_k(\theta_k^{(t)})\|^2 \leq \phi^2$), and $\mu_t$ is the lower bound on the global gradient norm (i.e., $\|\nabla\mathcal{L}(\bar{\theta}^{(t)})\| \geq \mu_t > 0$).*

**Practical Guidance.** Proposition 3 provides guidance for allocating communication to ensure $U^{(t)} \leq 0$, thereby guaranteeing the contribution of each step to the cumulative sum in Theorem 1. From Equation (11), note that $\phi$, $\sigma^2$, $\gamma^*$, $m$, and $L_q$ ($q \in \{1, \ldots, 4\}$) are time-independent, while the key time-varying factor is the gradient lower bound $\mu_t$, which tracks the optimization status of the averaged parameters $\bar{\theta}^{(t)}$ on the global landscape. Under this interpretation, Equation (11) implies that the communication-related term $p$ should be dynamically adjusted in response to the changing landscape geometry captured by $\mu_t$. Analytically, the left-hand side of Equation (11) is a strictly decreasing function of $p$, while the right-hand side is an increasing function of $\mu_t$. This implies a fundamental trade-off: more frequent communication (larger $p$) relaxes the condition, whereas a vanishing gradient (smaller $\mu_t$) tightens the allowable error bound. Specifically,

- **Early, high-gradient regime.** In the starting phase of training, when the globally averaged model is far from a minimum, the lower bound on gradient norm $\mu_t$ is large. This corresponds to a relaxed consensus requirement in Equation (11), which permits low-frequency communication (i.e., smaller $p$) without significantly impacting the performance of the globally merged model.

- **Late, low-gradient regime.** As models approach a solution and training enters a convergence phase, the gradient norm $\mu_t$ decreases. This tightens the constraint in Equation (11). In this regime, frequent communication (i.e., larger $p$) becomes critical.

We note that this theoretically motivated guidance aligns well with our empirical findings in Section 4 that more communication should be concentrated in the later stages of training.

## 6 IMPLICATIONS AND DISCUSSIONS

**Model Merging.** Our findings suggest potential implications for model merging. Recent work has shown that pretrained models occupy a flat "basic capability basin", while fine-tuning creates smaller "specific capability basins" (Chen et al., 2025). The observed "mergeability" suggests that decentralized learning may guide agents into connected specific capability basins, enabling simple permutation-free merging. This points to a practical direction: lightweight synchronization during local training may improve basin connectivity and simplify later merging into a stronger model.

**Decentralized Learning.** Our empirical and theoretical results show that decentralized learning can generalize under high data heterogeneity and limited communication, motivating adaptive algorithms that allocate communication budgets by monitoring training dynamics to satisfy the critical consensus edge condition in Equation (11).

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

## LLM Usage Statement

We use large language models (LLMs) as writing-assistance tools. Their role is confined to proof-reading and language polishing.

## Impact Statement

This paper studies the problem of temporal communication allocation in decentralized distributed learning, a highly significant topic in the era of communication-intensive large model training. Specifically, we aim to contribute to the development of communication-efficient decentralized learning without compromising performance. The potential positive social impacts are twofold:

- **Democratizing Access.** For individuals and organizations with constrained infrastructure, our work contributes to the democratization of access to large-scale collaborative training. By reducing communication requirements, we lower the barrier to entry for participating in advanced model development. Such inclusivity can extend the applicability of distributed learning systems to edge environments, thereby promoting more equitable contributions to models trained at scale.

- **Reducing Training Costs.** In datacenter environments, our approach can alleviate communication bottlenecks in distributed training. This reduction directly translates to shorter total wall-clock training time, thereby lowering the overall costs and energy consumption associated with large-scale distributed training.

No negative societal impacts are identified.

## Ethics Statement

Our research strictly adheres to the ICLR Code of Ethics. The work is foundational, focusing on the algorithmic and theoretical properties of decentralized learning, and does not involve human subjects or the collection of new sensitive data. All experiments were conducted on publicly available, standard academic datasets. We foresee no direct negative societal impacts; on the contrary, by reducing communication overhead, our findings may contribute positively by democratizing access to large-scale distributed training and lowering the associated resource footprint.

## Reproducibility Statement

We are committed to the reproducibility of our research. Our theoretical claims, including all assumptions and their justifications, are presented in Section 5 with complete, step-by-step proofs provided in Appendix D. Comprehensive details for reproducing our empirical results, including model architectures, data processing, hyperparameter settings, and communication configurations, are well documented in Appendix C.1.

## A    Limitations and Potential Questions

> *Q: Why use decentralized **AdamW** in some experiments when the theory is on decentralized **SGD**?*

**A**: We use decentralized AdamW in some of our experiments for its superior performance in non-IID settings. Crucially, all reported empirical observations remain fully consistent when using decentralized SGD, directly aligning with our theoretical analysis (see Figure 1 and Subsection C.3).

> *Q: How does the theory explain local models in decentralized learning are globally mergeable?*

**A**: The theoretical explanation for the "mergeability" of local models in decentralized learning is supported by our result that a globally merged model converges faster to the optimum than individual local models. Specifically, we provide a fine-grained convergence analysis showing that the model merged from decentralized SGD (DSGD) can match the convergence rate of parallel SGD, despite

limited communication. Since the rate of $m$-agent parallel SGD is superior to that of a single local model, this result transitively justifies the merged model's superior performance relative to any individual model, thereby providing theoretical support for mergeability.

---

**Q (Hyperparameter Tuning):** *How were the baselines tuned in terms of hyperparameters?*

---

**A:** All hyperparameters were tuned via grid search based on global test performance, with the batch size searched over $\{64, 128\}$. For ResNet-18 trained from scratch on Tiny ImageNet, we searched the learning rate over $\{1 \times 10^{-4}, 5 \times 10^{-4}, 1 \times 10^{-3}\}$ for AdamW and $\{1 \times 10^{-3}, 5 \times 10^{-3}, 1 \times 10^{-2}\}$ for SGD. For CLIP ViT-B/32 on Tiny ImageNet, we searched the learning rate over $\{1 \times 10^{-4}, 5 \times 10^{-4}, 1 \times 10^{-3}\}$ for AdamW and $\{5 \times 10^{-4}, 1 \times 10^{-3}, 5 \times 10^{-3}, 1 \times 10^{-2}\}$ for SGD. For the optimal hyperparameters selected for our main experiments, please refer to the **Implementation Details** in Appendix C.1 and the additional empirical results in Subsection C.3.

---

**Q (Comparison with Model Soup):** *How do initialization schemes affect results? Performance gains from merging have been observed in Model Soup (Wortsman et al., 2022a).*

---

**A:** We use different initialization schemes and observe consistent performance gains from global merging, whether models start from different random initializations or from a pretrained state. The majority of our experiments use different initializations, demonstrating that local models in decentralized learning can be effectively merged regardless of their starting points. This is quite surprising, as it contrasts with methods like **Model Soup**, which require models to be fine-tuned from an identical pretrained state. Furthermore, our experiments with a shared pretrained state confirm that the performance gains hold in that setting as well (see Figure 1a and Subsection C.3).

---

**Q (Methodology for Landscape Visualization):** *Please clarify the methodology for visualizing the loss landscape in Figure 1c, including the basis for the visualization grid.*

---

**A:** We adopt the visualization tool from (Crisostomi et al., 2024), positioning 16 trained models at the vertices of a regular hexadecagon. Any point within this polygon is an interpolated model whose parameters are determined by Wachspress barycentric coordinates; we then evaluate its cross-entropy loss to generate the contour map. Unlike methods that use random directions, our visualization grid is **deterministically** defined by the models themselves, allowing a direct investigation of their geometric connectivity. The full implementation is available in their official code repository https://github.com/crisostomi/cycle-consistent-model-merging/blob/master/notebooks/plots/plot_loss_contours_n_models.ipynb.

---

**Q (Experimental Scope):** *The empirical findings are restricted to visual tasks.*

---

**A:** Our empirical findings primarily focus on tasks within the vision domain. We note that this is consistent with most existing decentralized learning literature (Lin et al., 2021; Kong et al., 2021; Ying et al., 2021; Vogels et al., 2021; Li et al., 2022b; Zehtabi et al., 2025). Extending the experimental setup to broader tasks is a meaningful direction for future research.

---

**Q:** *The finding in Figure 2 (c), that local models eventually converge to a similar state even with limited communication, was also observed in prior work (Jelasity et al., 2005).*

---

**A:** In our setting, the local models do not, in fact, converge to a similar state or a single consensus point. This is because our work addresses a more challenging heterogeneous data regime, which differs from the setting in the cited prior work. Instead, we identify an emergent geometric structure where decentralized training guides local models to a shared "high-loss ring" surrounding a central low-loss basin (see Figure 1c). Although the models do not reach a consensus, they remain surprisingly mergeable within this region. This geometric arrangement allows their average, i.e., the globally merged model, to fall directly into the low-loss basin. To the best of our knowledge, we are the first to identify this emergent phenomenon in decentralized learning.

## B ADDITIONAL BACKGROUND AND RELATED WORK

### B.1 DECENTRALIZED LEARNING

Modern large-scale model training and inference are predominantly conducted within centralized, high-cost datacenters. Driven by mounting constraints on computational resources and power availability (Pilz et al., 2025), frontier-model training is already moving beyond a single datacenter, with Gemini 2.5 reporting synchronous data-parallel training across multiple TPUv5p pods distributed across multiple datacenters (Gemini Team, 2025). The decentralized-learning paradigm, drawing inspiration from swarm intelligence systems (Bonabeau et al., 1999; Mavrovouniotis et al., 2017), offers a more economical and scalable approach by distributing computational tasks across globally distributed nodes, rather than relying solely on a single central server (Yuan et al., 2022; Borzunov et al., 2023b; Jaghouar et al., 2024; Ramasinghe et al., 2025).

To provide context, we summarize key algorithmic and theoretical advances in decentralized learning. While our discussion highlights several notable contributions, it is not exhaustive; readers are referred to recent advances and surveys (Zhu et al., 2025; Martínez Beltrán et al., 2023; Singh et al., 2024; Yuan et al., 2024; He et al., 2025; Ramasinghe et al., 2025; Kolehmainen et al., 2025).

**Algorithmic Progress in Decentralized Learning**. The advancement of decentralized learning algorithms has been primarily driven by the need for communication efficiency in practical distributed learning. Decentralized algorithms have been refined to handle a variety of realistic scenarios, including time-varying communication topologies (Nedić & Olshevsky, 2015; Koloskova et al., 2020; Ying et al., 2021; Takezawa et al., 2023), asynchronous updates (Lian et al., 2018; Xu et al., 2021; Nadiradze et al., 2021; Bornstein et al., 2023; Even et al., 2024), statistical heterogeneity (Tang et al., 2018; Vogels et al., 2021; Le Bars et al., 2023), and robustness to Byzantine failures (He et al., 2022; Ye & Ling, 2025). Moreover, recent work has extended beyond standard empirical risk minimization to more structured problem classes, such as compositional (Gao & Huang, 2021), minimax (Xian et al., 2021; Zhu et al., 2023a; Chen et al., 2024), and bi-level optimization (Yang et al., 2022; Gao et al., 2023; Chen et al., 2023). Additionally, privacy concerns in decentralized learning are also critical, with efforts focusing on differential privacy (Cyffers et al., 2024; Allouah et al., 2024) and data reconstruction attacks (Mrini et al., 2024).

**Theoretical Progress in Decentralized Learning.** Foundational work on decentralized optimization (Nedic & Ozdaglar, 2009; Sayed, 2014; Yuan et al., 2016; Lian et al., 2017) laid the groundwork for understanding convergence. Building on this, Lu & De Sa (2021) proposed a hierarchical abstraction of decentralization, decomposing it into three layers, providing a unified view across federated and decentralized paradigms. Koloskova et al. (2020) consolidated synchronous decentralized SGD algorithms with changing communication topologies and local updates, and Even et al. (2024) extended the unifying perspective to asynchronous protocols. More recently, Zehtabi et al. (2025) developed these frameworks further by considering the sporadicity of both communication and computations. On the generalization front, Richards et al. (2020) derived stability-based bounds for decentralized SGD in convex settings, while Sun et al. (2021) extended these to non-convex objectives, revealing a dependency on the spectral gap of the communication graph. This dependency was subsequently refined by Zhu et al. (2022), who introduced a Gaussian weight difference assumption to tighten the bound. Complementary results showed that in convex regimes, the generalization of decentralized SGD matches that of centralized SGD (Le Bars et al., 2024), while in non-convex landscapes, decentralization primarily impacts worst-case generalization behavior. To account for unexplained generalization behaviors in decentralized training (Kong et al., 2021; Gurbuzbalaban et al., 2022; Vogels et al., 2023), Zhu et al. (2023b) linked decentralized SGD to random sharpness-aware minimization (SAM), revealing a bias toward flatter minima. Notably, akin to our finding that decentralized learning generalizes when allocated high communication late in training, Zhou et al. (2025) has shown that SAM efficiently selects flatter minima in the later stages of training.

**Towards Decentralized Training of Foundation Models.** Recent advances have shown the feasibility of training large-scale foundation models in decentralized environments. DT-FM (Yuan et al., 2022) introduced tasklet-based scheduling for Transformer training under bandwidth-constrained settings, enabling efficient resource allocation. SWARM Parallelism (Ryabinin et al., 2023) scaled decentralized training through resilient pipeline design and adaptive load balancing. CocktailSGD (Wang et al., 2023) further improved efficiency via a combination of decentralization, gradient sparsification, and quantization for LLM fine-tuning. On the inference side, Petal (Borzunov et al., 2023a)

exploited peer-to-peer networks to amortize computational costs across heterogeneous nodes. Most recently, Intellect (Jaghouar et al., 2024), building on Diloco (Douillard et al., 2023), leveraged hybrid parallelism, i.e., both data and model parallelism, to collaboratively train models with billions of parameters. NoLoCo (Kolehmainen et al., 2025) further extended Diloco to gossip-type decentralized settings. For a broad survey of large-scale deep learning practice, see Shen et al. (2024; 2025).

### B.2 Implicit Bias of Decentralized Learning

The concept of implicit bias, i.e., the intrinsic preference of learning algorithms for solutions with certain properties, has emerged as a key concept in explaining the empirical success of modern deep learning (Li et al., 2022c; Vardi, 2023; Lyu, 2024). Recent studies have highlighted intriguing distinctions between decentralized stochastic gradient descent (DSGD) and its centralized counterpart (CSGD). Gurbuzbalaban et al. (2022) demonstrated that under certain conditions, DSGD operating on large, sparse topologies exhibits heavier-tailed parameter distributions compared to CSGD. Zhang et al. (2021) showed that decentralization introduces landscape-dependent noise, which can improve tolerance to larger learning rates. This observation aligns with findings by Vogels et al. (2023), who revealed that collaboration in decentralized settings permits the use of larger learning rates. Zhu et al. (2023b) first explicitly characterized the implicit bias of decentralized SGD by establishing its connection with random sharpness-aware minimization, proving the existence of flatness bias in decentralized training. Complementing this, Cao et al. (2024) offered a detailed analysis of the interplay between flatness and optimization in DSGD, particularly its ability to escape local minima. More recently, Wu & Sun (2024) investigated the implicit regularization properties of decentralized optimization in non-convex sparse regression problems, recovering the convergence rates achieved by gradient descent in centralized settings.

**Comparison with Zhu et al. (2023b).** We note that Zhu et al. (2023b) has highlighted the generalization benefits of decentralized learning, but key differences exist in terms of the experimental setup and the insights derived. While Zhu et al. (2023b) focused on IID scenarios and specific cases involving exceptionally large batch sizes, we consider the more realistic non-IID setting using standard batch sizes. This shift in focus allows us to uncover phenomena not observed by Zhu et al. (2023b), including insights into communication allocation strategies.

### B.3 Model Merging

**Mode Connectivity and Model Merging Techniques.** Recent works on *(Linear) Mode Connectivity* have advanced our understanding of the complex loss landscape in neural networks. Freeman & Bruna (2017); Draxler et al. (2018); Garipov et al. (2018); Nagarajan & Kolter (2019); Frankle et al. (2020) discovered that different solutions of deep neural networks can be merged by simply averaging their parameters. Sonthalia et al. (2025) further showed that the solutions may form a star domain. We note that this phenomenon has been observed in the following scenarios:

- *Shared initialization (Frankle et al., 2020; Fort et al., 2020; Zhou et al., 2023).* Models are initialized from a pretrained checkpoint.

- *Homogeneous data distribution (Wortsman et al., 2022a).* Models are trained on homogeneous data distribution.

- *Permutation (Ainsworth et al., 2023; Entezari et al., 2022).* Models are independently trained. The neurons of one model are permuted to match the neurons of the other while maintaining a functionally equivalent network.

These findings have inspired a range of model merging techniques for various applications. Izmailov et al. (2018); Matena & Raffel (2022); Rame et al. (2022; 2023); Wortsman et al. (2022a;b) found that merging the parameters of models that start from the same pretrained model and are fine-tuned on the same task leads to improved generalization and robustness. Furthermore, Ilharco et al. (2022); Li et al. (2022a); Ilharco et al. (2023); Ortiz-Jimenez et al. (2023); Yadav et al. (2023) showed that merging models that are fine-tuned on different tasks enables multitask capabilities.

**Comparisons with Model Merging Literature.** Our results show that mode connectivity, or mergeability, can still emerge in decentralized learning, even when the local models are initialized *differently*, trained on highly *heterogeneous* data, and merged *without* any permutation. Our findings

offer new insights into both model merging techniques and the geometry of the neural network loss landscape, which we anticipate will motivate further advances in both areas.

## C  ADDITIONAL EXPERIMENTS

### C.1  EXPERIMENTAL SETUPS

**Computational Resources.** The experiments were conducted on a computing facility equipped with 80 GB NVIDIA® A100™ GPUs. All implementations are based on PyTorch, and computations are distributed across multiple GPUs for efficiency.

**Dataset.** We use three widely adopted image classification datasets: CIFAR-10, CIFAR-100 (Krizhevsky et al., 2009), and Tiny ImageNet (Le & Yang, 2015). CIFAR-10 consists of 60,000 RGB images across 10 classes, while CIFAR-100 contains 60,000 RGB images across 100 classes. The images in both datasets have a spatial resolution of $32 \times 32$ pixels. Tiny ImageNet is a subset of the ImageNet dataset, comprising 100,000 images drawn from 200 classes, with each image resized to $64 \times 64$ pixels. It provides a mid-scale benchmark that is more challenging than CIFAR datasets but less computationally demanding than training on the full ImageNet dataset. To incorporate data augmentation, we employ a combination of RandomCrop with 4-pixel padding, RandomHorizontalFlip, and RandAugment with `num_ops`=2 and `magnitude`=9.

**Details of Decentralized Learning**. We simulate a heterogeneous decentralized learning environment. For our main experiments (Figure 1a and Figure 1b), we use $m = 32$ agents, while for other experiments, including the sliding window experiments (Figure 2) and the loss landscape visualizations (Figure 1c), we use $m = 16$ agents. The number of agents for the visualization was chosen as 16 for clarity, as a plot with 32 models would be visually crowded. In all configurations, we employ a Dirichlet distribution characterized by $\alpha = 0.1$ to partition the data among agents. The Dirichlet distribution is commonly used to partition data in federated learning scenarios, as it controls label-distribution skew among agents (Yurochkin et al., 2019; Hsu et al., 2019). A smaller $\alpha$ results in more imbalanced data distributions, where some agents predominantly receive data from a limited number of classes, while a larger $\alpha$ results in more uniform label distributions across agents. This configuration effectively captures the realistic non-IID nature of decentralized learning, where different agents may have access to personalized data reflective of their local environments.

- **Communication Graph.** We evaluate three decentralized communication topologies: random graph, ring graph, and exponential graph. In the random graph setting, during each communication round, each agent selects a random subset of its neighbors for gossip averaging. For "R 1", each agent selects exactly one random neighbor in each round. For "R 0.2", each agent selects one neighbor with probability 0.2 and continues local training without communication with probability 0.8. The ring graph enforces a fixed cyclic communication structure, while the exponential graph ensures connectivity by allowing agents to communicate at exponentially increasing distances.

- **Communication Rounds and Local Steps.** The decentralized learning process is conducted over $T = 300$ communication rounds. We use $H = 100$ local minibatch steps per communication round to balance communication and computation costs.

- **Local Data per Agent.** Each agent is assigned a subset of the dataset with a fixed size of $4096$ samples, drawn according to a Dirichlet distribution to simulate realistic non-IID scenarios.

**Model Architecture.** To ensure a representative comparison across different model families, we adopt ResNet-18 (He et al., 2016) and CLIP ViT-B/32 (Radford et al., 2021) as backbone architectures in our experiments. ResNet-18 is a widely used lightweight convolutional neural network that serves as a canonical example of traditional CNN-based architectures. In contrast, CLIP ViT-B/32 is a transformer-based vision model pretrained on large-scale image–text pairs. For experiments on Tiny ImageNet, where images are resized to 64×64 pixels, we adjust the CLIP visual encoder to handle the lower resolution. With a patch size of 32, each image yields 4 visual tokens arranged in a 2×2 grid, plus a [CLS] token, resulting in a 5-token input sequence.

**Implementation Details.** All hyperparameters are tuned by grid search based on global test performance (see Definition 1). For experiments using decentralized SGD, the optimal learning rates were found to be $1 \times 10^{-2}$ for ResNet-18 (trained from scratch) and $1 \times 10^{-3}$ for CLIP ViT-B/32. When using decentralized AdamW, the optimal learning rate is $5 \times 10^{-4}$ for ResNet-18 (both when trained

from scratch and fine-tuned from ImageNet-pretrained weights) and $1 \times 10^{-5}$ for the pretrained CLIP ViT-B/32 on Tiny ImageNet. For all experiments, weight decay is set to $5 \times 10^{-4}$ and the batch size is selected as 128. The key empirical results remain consistent across these optimizer and hyperparameter choices, indicating that our conclusions are stable and not sensitive to specific hyperparameter configurations.

**Details of Loss Landscape Visualization in Figure 1c.** To analyze the geometric connections among models after decentralized training, we visualize the loss landscape spanning their parameter spaces. We adopt the visualization tool from (Crisostomi et al., 2024), which is specifically designed to analyze the interpolation space within the convex hull formed by a given set of models. In our implementation, we position the 16 trained models at the vertices of a regular hexadecagon. Any point within this polygon represents an interpolated model, whose parameters are a weighted sum of the parameters of the 16 vertex models; the weights are determined by the point's Wachspress barycentric coordinates. We then evaluate the cross-entropy loss of each interpolated model on the entire test set to generate the final loss contour map, as shown in Figure 1c. The implementation is available in their notebook https://github.com/crisostomi/cycle-consistent-model-merging/blob/master/notebooks/plots/plot_loss_contours_n_models.ipynb within the official code repository for (Crisostomi et al., 2024). We note two key aspects of this visualization approach:

- **Focus on Convex Combinations.** For points outside the polygon, one or more of their barycentric coordinates become negative, corresponding to an extrapolation, which is often unstable. This visualization approach is consistent with Definition 2, focusing on the space of convex combinations among the models.

- **Deterministic Grid vs. Random Directions.** Notably, the visualization method differs from approaches that use random directions to probe the landscape of a single model, as our visualization grid is defined directly by the 16 models themselves. This allows us to directly investigate the geometric connectivity and interpolation properties among this predefined set of models.

**Computational Resource Requirements and Runtime.** To enhance accessibility, our code includes a centralized simulation of decentralized training. This enables the reproduction and extension of our decentralized learning experiments using fewer GPUs. A single decentralized AdamW training experiment with 16 agents using ResNet-18 on the Tiny ImageNet dataset requires approximately 15 GB of GPU memory and can be conducted on a single GPU with sufficient memory, such as an NVIDIA V100, RTX 3090, RTX 4090, or A100. On an A100 GPU, the typical runtime is approximately 8 hours for 300 communication rounds, each comprising 100 local steps. For the CLIP ViT-B/32 model, the memory demand rises to about 30 GB, yet it remains feasible on a single A100 GPU, with a runtime of approximately 12 hours under the same configuration of 300 communication rounds and 100 local steps per round.

## C.2 PRACTICAL EVALUATION METRICS

The standard evaluation metric for parallel and federated learning is the accuracy of the global model.

**Definition C.1** (Test Accuracy of Global Model). *The accuracy of the global model $\theta$ is defined as:*

$$\text{Acc}(\theta) \triangleq \frac{1}{m} \sum_{k \in \mathcal{V}} \mathbb{E}_{\xi_k \sim \mathcal{D}_k} \text{Acc}(\theta; \xi_k) \overset{\text{if IID}}{=} \mathbb{E}_{\xi \sim \mathcal{D}} \text{Acc}(\theta; \xi).$$

In decentralized learning, models are often evaluated in the absence of a full consensus model $\theta$ due to data heterogeneity and limited training time. Two main metrics are used in this scenario.

**Definition C.2** (Average Local Test Accuracy). *The average accuracy of agents $k \in \mathcal{V}$ is defined as:*

$$\underbrace{\overline{\text{Acc}}(\{\theta_k\}_{k \in \mathcal{V}}) \triangleq \frac{1}{m} \sum_{k \in \mathcal{V}} \mathbb{E}_{\xi_k \sim \mathcal{D}_k} \text{Acc}(\theta_k; \xi_k)}_{\textit{Average Test accuracy on the local distribution across agents}} \overset{\text{if IID}}{=} \frac{1}{m} \sum_{k \in \mathcal{V}} \mathbb{E}_{\xi \sim \mathcal{D}} \text{Acc}(\theta_k; \xi).$$

> **Remark C.1** (Local Generalization). This metric aims to address the following question in decentralized learning: *how well do local models $\{\theta_k\}_{k \in \mathcal{V}}$, with the aid of peer-to-peer communication, generalize to their local (personalized) data distribution $\mathcal{D}_l$?* This is the standard evaluation metric in personalized decentralized settings, where the goal is to optimize local objectives.

However, in real-world scenarios, local data distributions are often heterogeneous and not guaranteed to be IID across agents. In such settings, an important goal is to understand how well local models, trained on limited local data, generalize to the global data distribution. To account for this, we adopt the following *average global test accuracy*, a proxy for the average global population risk, as the primary evaluation metric, which quantifies how well local models generalize to the global distribution.

**Definition C.3** (Average Global Test Accuracy). *The average accuracy of agents $k \in \mathcal{V}$ is defined as:*

$$\underbrace{\overline{\mathrm{Acc}}(\{\theta_k\}_{k \in \mathcal{V}}) = \frac{1}{m} \sum_{k \in \mathcal{V}} \mathrm{Acc}(\theta_k),}_{\textit{Average Accuracy across agents}} \quad where \ \mathrm{Acc}(\cdot) \triangleq \underbrace{\frac{1}{m} \sum_{l \in \mathcal{V}} \mathbb{E}_{\xi_l \sim \mathcal{D}_l} \mathrm{Acc}(\cdot; \xi_l)}_{\textit{Test accuracy on the global distribution}} \ .$$

**Remark C.2** (Global Generalization). This metric is specifically designed to address a core research question in fully decentralized learning with non-IID data: *how well do local models $\{\theta_k\}_{k \in \mathcal{V}}$, trained with limited peer-to-peer synchronization, generalize to the global data distribution $\mathcal{D}$?* We note that this objective is particularly critical in the highly non-IID scenarios we study, where local models drift significantly apart. Unlike federated learning that measures the performance of a global model, this metric offers a more realistic evaluation for decentralized settings where no central server is present.

## C.3 ADDITIONAL EXPERIMENTS

### C.3.1 DIFFERENT NUMBERS OF AGENTS AND OPTIMIZERS

We conduct additional experiments by varying the number of agents (from 16 to 32) and comparing different optimizers (from SGD to AdamW). The effect of single merging remains consistent.

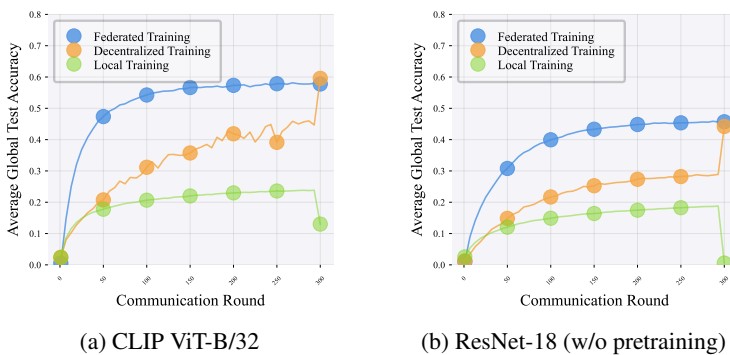

(a) CLIP ViT-B/32       (b) ResNet-18 (w/o pretraining)

Figure C.1: **(a, b)**: Global test accuracy (see Definition 1) of CLIP ViT-B/32 (a) and ResNet-18 (b) trained on Tiny ImageNet using FedAdamW (blue), decentralized AdamW (orange), and one-shot FedAdamW (green), distributed across **16** agents with high data heterogeneity (Dirichlet $\alpha = 0.1$). Decentralized training involves each agent syncing parameters with a random peer in each round with probability 0.2, with a single global merging at the final round (see details in Appendix C.1).

### C.3.2 DIFFERENT COMMUNICATION TOPOLOGIES

We also conduct additional experiments with different communication topologies to examine whether the empirical results remain consistent. The observations are summarized below.

• **Models remain mergeable under different numbers of peers.** We evaluate two settings (random topology with $R = 0.2$ and $R = 1$; see "Communication Graph" in Appendix C.1). Figure C.3a shows consistent performance improvements, with more significant gains in the $R = 0.2$ case.

• **Models remain mergeable across different communication topologies.** We evaluate two topologies: exponential and ring graphs. As shown in Figure C.3b, both topologies preserve the mergeability

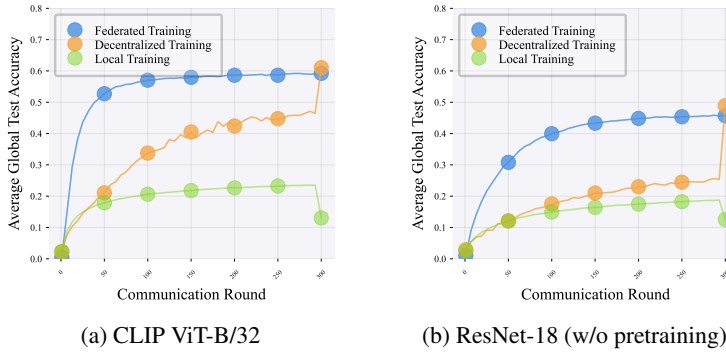

(a) CLIP ViT-B/32         (b) ResNet-18 (w/o pretraining)

Figure C.2: **(a, b)**: Global test accuracy (see Definition 1) of CLIP ViT-B/32 (a) and ResNet-18 (b) trained on Tiny ImageNet using FedAdamW (blue), decentralized AdamW (orange), and one-shot FedAdamW (green), distributed across **32** agents with high data heterogeneity (Dirichlet $\alpha = 0.1$). Decentralized training involves each agent syncing parameters with a random peer in each round with probability 0.2, with a single global merging at the final round (see details in Appendix C.1).

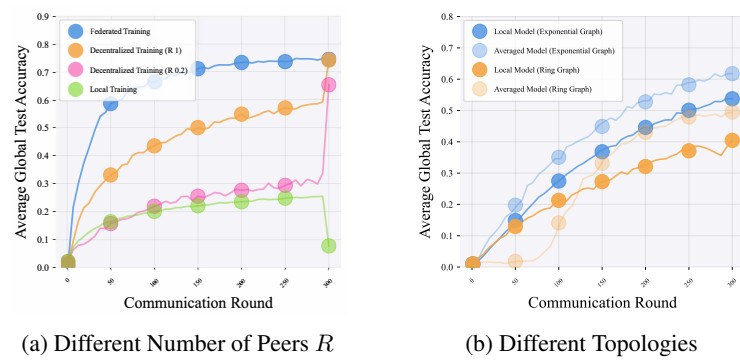

(a) Different Number of Peers $R$        (b) Different Topologies

Figure C.3: Global test accuracy (see Definition 1) of training ResNet-18 on Tiny ImageNet, distributed across 16 agents with high heterogeneity (Dirichlet $\alpha = 0.1$; see details in Appendix C.1). We evaluate the effects of different **(a)** numbers of peers $R$, and **(b)** communication topologies. Pretrained weights are used only in (a).

of local models, with exponential graphs yielding slightly better test performance for both local and merged models. The trend of mergeability persists across topologies throughout training, though performance may vary.

### C.3.3 DIFFERENT HYPERPARAMETERS, DATASET, AND HETEROGENEITY LEVEL

We further conduct supplementary experiments in which the final global merge is approximated by topology-constrained gossip merging on an exponential graph (Ying et al., 2021).

**Summary**. We observe consistent test-performance improvements from a single global merging across a wide range of settings, including different hyperparameter setups, datasets, degrees of data heterogeneity, model architectures, optimizers, initialization schemes, and communication topologies.

### C.3.4 REALIZING FINAL GLOBAL GOSSIP VIA TOPOLOGY-CONSTRAINED MERGING

We also perform additional experiments where the final global merging is approximated by multiple rounds of gossip merging using a specific exponential topology (Ying et al., 2021).

**Summary**. We observe that (1) even a single round of topology-constrained final gossip merging substantially improves global test accuracy, and (2) the resulting performance is comparable to the baseline that uses random communication among all agents followed by one perfect global merge.

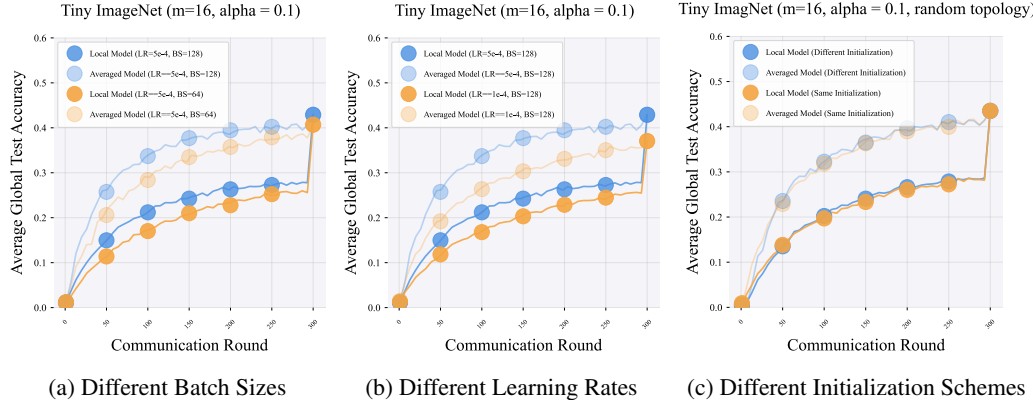

(a) Different Batch Sizes  (b) Different Learning Rates  (c) Different Initialization Schemes

Figure C.4: Global test accuracy (see Definition 1) of training ResNet-18 on Tiny ImageNet with decentralized AdamW, distributed across 16 agents with high heterogeneity (Dirichlet $\alpha = 0.1$; see details in Appendix C.1). We evaluate the effects of different **(a)** batch sizes (64 vs. 128), **(b)** learning rates ($5 \times 10^{-4}$ vs. $1 \times 10^{-4}$), and **(c)** different initialization schemes.

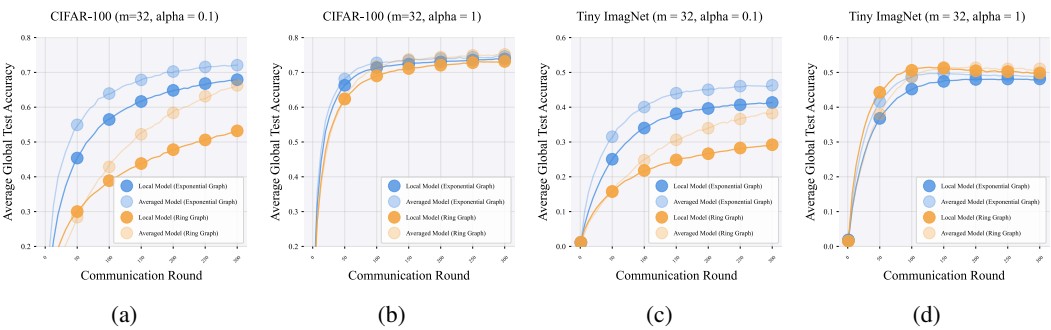

(a)  (b)  (c)  (d)

Figure C.5: Global test accuracy (see Definition 1) for ResNet-18 trained with decentralized AdamW across 32 agents under different levels of data heterogeneity (Dirichlet $\alpha = 0.1$ (**a, c**) vs. $\alpha = 1.0$ (**b, d**); see Appendix C.1). Results are reported on both CIFAR-100 (**a, b**) and Tiny ImageNet (**c, d**).

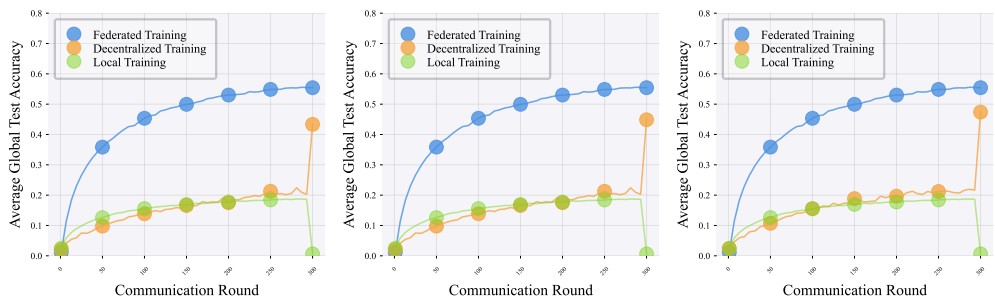

(a) 1 Round Final Gossip Merging (b) 5 Rounds Final Gossip Merging (c) 1 Round Final Global Merging

Figure C.6: Global test accuracy (see Definition 1) for ResNet-18 trained with decentralized AdamW on Tiny ImageNet (32 agents). Left: regular decentralized training followed by one round of topology-constrained final gossip merging on an exponential graph. Middle: the same topology-constrained setting, but with five rounds of final gossip merging to better approximate global aggregation. Right: baseline setting (our original approach), with random communication among all agents during training followed by one perfect global merge.

## D  THEORY

This section provides the proofs of the main theoretical results presented in this paper. For simplicity, and following the setup in the existing literature, we assume that the sample size of local agents is $n_k = n$ for all $k \in \mathcal{V}$.

---

**Lemma D.1** (Consensus Distance Recursion under Local Updates (Kong et al., 2021)). *Suppose Assumption 1–Assumption 3 hold. Let $\theta_k^{(t)}$ be the local parameter on client $k$ at the $t$-th step, and denote their average by $\bar{\theta}^{(t)} = \frac{1}{m} \sum_{k=1}^{m} \theta_k^{(t)}$. Define the* consensus distance *and the* average gradient norm *at round $t$ by $\Xi_t^2 = \frac{1}{m} \sum_{k=1}^{m} \|\theta_k^{(t)} - \bar{\theta}^{(t)}\|^2$ and $\phi_t^2 = \frac{1}{m} \sum_{k=1}^{m} \|\nabla \mathcal{L}_k(\theta_k^{(t)})\|^2$, where $\mathcal{L}_k(\theta) = \mathbb{E}_{\xi_k \sim \mathcal{D}_k}[\mathcal{L}(\theta; \xi_k)]$. Let $\eta > 0$ be the learning rate, and let $\sigma^2$ be the variance bound from Assumption 3. Then there exists a constant $p > 0$ (see Assumption 1) such that for all $t \geq 0$, the following inequality holds:*

$$\mathbb{E}\left[\Xi_{t+1}^2\right] \leq \left(1 - \frac{p}{2}\right) \Xi_t^2 + \frac{12(1-p)}{p} \eta^2 \left(\phi_t^2 + \sigma^2\right), \tag{D.1}$$

*where the expectation is taken over the stochastic gradients in the $t$-th update phase.*

---

*Proof.* For completeness, we provide the proof of Lemma D.1, with minor corrections and additional details. In decentralized SGD (Algorithm 1 with SGD as the local optimizer), each agent $k \in \mathcal{V}$ performs at each iteration

$$\theta_k^{(t+1)} = \sum_{l=1}^{m} W_{k,l} \left(\theta_l^{(t)} - \eta \nabla \mathcal{L}_l(\theta_l^{(t)}; \xi_l^{(t)})\right).$$

In matrix form, letting

$$\Theta^{(t)} = [\theta_1^{(t)}, \dots, \theta_m^{(t)}] \in \mathbb{R}^{d \times m}, \quad \nabla \mathcal{L}(\Theta^{(t)}; \xi^{(t)}) = [\nabla \mathcal{L}_1(\theta_1^{(t)}; \xi_1^{(t)}), \dots, \nabla \mathcal{L}_m(\theta_m^{(t)}; \xi_m^{(t)})],$$

we have

$$\Theta^{(t+1)} = \left(\Theta^{(t)} - \eta \nabla \mathcal{L}(\Theta^{(t)}; \xi^{(t)})\right) W.$$

The consensus matrix after mixing is

$$\bar{\Theta}^{(t+1)} = \Theta^{(t+1)} \frac{1}{m} \mathbf{1} \mathbf{1}^\top = \left(\Theta^{(t)} - \eta \nabla \mathcal{L}(\Theta^{(t)}; \xi^{(t)})\right) \frac{1}{m} \mathbf{1} \mathbf{1}^\top,$$

since $\mathbf{1}^\top W = \mathbf{1}^\top$.

Thus the consensus distance satisfies

$$m \Xi_{t+1}^2 = \left\|\Theta^{(t+1)} - \bar{\Theta}^{(t+1)}\right\|_F^2 = \left\|\left(\Theta^{(t)} - \eta \nabla \mathcal{L}(\Theta^{(t)}; \xi^{(t)})\right)\left(W - \frac{1}{m} \mathbf{1} \mathbf{1}^\top\right)\right\|_F^2.$$

By Assumption 1, for any $\Theta \in \mathbb{R}^{d \times m}$,

$$\mathbb{E}_W \|\Theta W - \bar{\Theta}\|_F^2 \leq (1 - \rho) \|\Theta - \bar{\Theta}\|_F^2,$$

we obtain

$$m \Xi_{t+1}^2 \leq (1 - p) \left\|\Theta^{(t)}\left(I - \frac{1}{m} \mathbf{1} \mathbf{1}^\top\right) - \eta \nabla \mathcal{L}(\Theta^{(t)}; \xi^{(t)})\left(I - \frac{1}{m} \mathbf{1} \mathbf{1}^\top\right)\right\|_F^2.$$

Applying the inequality $\|A + B\|_F^2 \leq (1 + \alpha)\|A\|_F^2 + (1 + 1/\alpha)\|B\|_F^2$ with $\alpha = \frac{p}{2}$ gives

$$m \Xi_{t+1}^2 \leq (1 - p) \left[\left(1 + \tfrac{p}{2}\right) \left\|\Theta^{(t)}\left(I - \tfrac{1}{m} \mathbf{1} \mathbf{1}^\top\right)\right\|_F^2 + \left(1 + \tfrac{2}{p}\right) \eta^2 \left\|\nabla \mathcal{L}(\Theta^{(t)}; \xi^{(t)})\right\|_F^2\right]$$

$$\leq \left(1 - \tfrac{p}{2}\right) m \Xi_t^2 + \frac{6(1-p)}{p} \eta^2 \left\|\nabla \mathcal{L}(\Theta^{(t)}; \xi^{(t)})\right\|_F^2,$$

where we use $(1 + p/2) \leq 1 + p$ and $(1 + 2/p) \leq 6/p$ for $p \in (0, 1)$.

We now decompose the stochastic gradient as

$$\nabla\mathcal{L}(\Theta^{(t)};\xi^{(t)}) = \nabla\mathcal{L}(\Theta^{(t)}) + \left[\nabla\mathcal{L}(\Theta^{(t)};\xi^{(t)}) - \nabla\mathcal{L}(\Theta^{(t)})\right],$$

so by Young's Inequality, we have

$$\left\|\nabla\mathcal{L}(\Theta^{(t)};\xi^{(t)})\right\|_F^2 \le 2\left\|\nabla\mathcal{L}(\Theta^{(t)})\right\|_F^2 + 2\left\|\nabla\mathcal{L}(\Theta^{(t)};\xi^{(t)}) - \nabla\mathcal{L}(\Theta^{(t)})\right\|_F^2.$$

Taking expectation over $\xi^{(t)}$ and invoking Assumption 3, we get

$$\mathbb{E}\left[\|\nabla\mathcal{L}(\Theta^{(t)};\xi^{(t)})\|_F^2\right] \le 2\|\nabla\mathcal{L}(\Theta^{(t)})\|_F^2 + 2\,\sigma^2 m.$$

Substituting back and dividing by $m$ yields

$$\mathbb{E}\left[\Xi_{t+1}^2\right] \le \left(1 - \tfrac{p}{2}\right)\Xi_t^2 \;+\; \frac{12(1-p)}{p}\,\eta^2\big(\phi_t^2 + \sigma^2\big),$$

which completes the proof. $\qquad\square$

---

**Corollary D.2** (Upper Bounds of Consensus Distance (Kong et al., 2021)). *Define the* consensus distance *and the* average gradient norm *at round $t$ by $\Xi_t^2 = \frac{1}{m}\sum_{k=1}^m \|\theta_k^{(t)} - \bar{\theta}^{(t)}\|^2$ and $\phi_t^2 = \frac{1}{m}\sum_{k=1}^m \|\nabla\mathcal{L}_k(\theta_k^{(t)})\|^2$, where $\mathcal{L}_k(\theta) = \mathbb{E}_{\xi_k\sim\mathcal{D}_k}[\mathcal{L}(\theta;\xi_k)]$. Under the conditions of Lemma D.1, suppose that for all iterations $t$, the gradient norms are uniformly bounded by a constant $\phi$, i.e. $\phi_t^2 \le \phi^2$, $\forall t \in \{1,\dots,T\}$. Then the expected consensus distance satisfies*

$$\mathbb{E}\left[\Xi_t^2\right] \;\le\; \frac{24\,(1-p)\,\eta^2}{p^2}\Big(\phi^2 + \sigma^2\Big).$$

*In the general case where the gradient norms change slowly, i.e., $\phi_t^2 \le (1 + \tfrac{p}{4})\,\phi_{t+1}^2$, we have*

$$\mathbb{E}\left[\Xi_t^2\right] \;\le\; \frac{48\,(1-p)\eta^2}{p^2}\left(\phi_{t-1}^2 + \sigma^2\right).$$

*The expectation here is taken over the stochastic gradients in the $t$-th update phase.*

---

*Proof.* Consider the key recursion from Lemma D.1:

$$\mathbb{E}\left[\Xi_{t+1}^2\right] \;\le\; \left(1 - \tfrac{p}{2}\right)\Xi_t^2 \;+\; \frac{12\,(1-p)}{p}\,\eta^2\big(\phi_t^2 + \sigma^2\big).$$

**(1) Special Case: uniformly bounded gradient norms.**

Assume $\phi_t^2 \le \phi^2$. Unrolling the above gives

$$\mathbb{E}\left[\Xi_{t+1}^2\right] \;\le\; \sum_{i=0}^{t-1}\left(1 - \tfrac{p}{2}\right)^i \frac{12\,(1-p)}{p}\,\eta^2\left(\phi^2 + \sigma^2\right).$$

Since $\sum_{i=0}^{t-1}(1 - \tfrac{p}{2})^i \le \tfrac{2}{p}$, we can bound the consensus distance as

$$\mathbb{E}\left[\Xi_{t+1}^2\right] \;\le\; \frac{12\,(1-p)}{p}\,\eta^2(\phi^2 + \sigma^2) \;\times\; \frac{2}{p} \;=\; \frac{24\,(1-p)\,\eta^2}{p^2}\,(\phi^2 + \sigma^2),$$

which yields the first claim.

**(2) Special Case: slowly changing gradient norms.**

If $\phi_t^2 \le (1 + \tfrac{p}{4})\,\phi_{t+1}^2$, and since

$$\left(1 - \tfrac{p}{2}\right)^i\left(1 + \tfrac{p}{4}\right)^i \;\le\; \left(1 - \tfrac{p}{4}\right)^i,$$

the consensus distance satisfies

$$\mathbb{E}\left[\Xi_{t+1}^2\right] \le \sum_{i=0}^{t-i-1}\left(1-\frac{p}{2}\right)^i \frac{12(1-p)\eta^2\left(\phi_{t-1}^2+\sigma^2\right)}{p}$$

$$\le \sum_{i=0}^{t-1}\left(1-\frac{p}{4}\right)^i \frac{12(1-p)\eta^2(\phi_{t-1}^2+\sigma^2)}{p} \le \frac{48(1-p)\eta^2}{p^2}\left(\phi_{t-1}^2+\sigma^2\right). \tag{D.2}$$

$\square$

**Proposition D.3** (Implicit Bias of Decentralized SGD (Zhu et al., 2023b)). *Suppose Assumption 2 holds, the globally averaged model of decentralized SGD (DSGD), defined by $\bar{\theta}^{(t)} = \frac{1}{m}\sum_{k=1}^m \theta_k^{(t)}$, follows the following gradient descent direction:*

$$\mathbb{E}_{\xi^{(t)}}[\bar{\theta}^{(t+1)}] = \bar{\theta}^{(t)} - \eta \cdot \mathbb{E}_{\epsilon^{(t)}\sim\mathcal{N}(0,\Gamma^{(t)})}\left[\nabla\mathcal{L}(\bar{\theta}^{(t)}+\epsilon^{(t)})\right] + \delta^{(t)},$$

*where $\Gamma^{(t)} = \frac{1}{m}\sum_{k=1}^m (\theta_k^{(t)}-\bar{\theta}^{(t)})(\theta_k^{(t)}-\bar{\theta}^{(t)})^\top \in \mathbb{R}^{m\times m}$ denotes the consensus distance matrix, and $\delta^{(t)} = O\left(\frac{\eta}{m}\sum_{k=1}^m \|\theta_k^{(t)}-\bar{\theta}^{(t)}\|_2^3\right)$ denotes the higher-order terms. The first expectation eliminates the randomness from sampled data $\xi^{(t)} = \{\xi_k^{(t)}\}_{k\in\mathcal{V}}$ at step $(t)$.*

We can then control the expected squared distance between two consecutive steps of the globally averaged model with Corollary D.4.

**Corollary D.4.** *Under the assumptions in Proposition D.3, the expected squared distance between two consecutive iterates of decentralized SGD can be bounded as follows:*

$$\mathbb{E}_{\xi^{(t)}}\left\|\bar{\theta}^{(t+1)}-\bar{\theta}^{(t)}\right\|^2 \le \frac{\eta^2\sigma^2}{m} + \eta^2\left\|\nabla\mathcal{L}(\bar{\theta}^{(t)}) + \frac{1}{2}\nabla\operatorname{Tr}\left(\nabla^2\mathcal{L}(\bar{\theta}^{(t)})\Gamma^{(t)}\right)+\delta^{(t)}\right\|^2. \tag{D.3}$$

*Proof.* Denote $\gamma^{(t+1)} = \mathbb{E}_{\xi^{(t)}}\bar{\theta}^{(t+1)} - \bar{\theta}^{(t)}$. We can expand the expected distance as follows:

$$\mathbb{E}_{\xi^{(t)}}\left\|\bar{\theta}^{(t+1)}-\bar{\theta}^{(t)}\right\|^2 = \mathbb{E}_{\xi^{(t)}}\left\|\bar{\theta}^{(t+1)}\right\|^2 + \left\|\bar{\theta}^{(t)}\right\|^2 - 2(\bar{\theta}^{(t)})^\top\gamma^{(t+1)}$$

$$= \operatorname{Tr}\left(\operatorname{Cov}(\bar{\theta}^{(t+1)})\right) + \left\|\mathbb{E}_{\xi^{(t)}}\bar{\theta}^{(t+1)}\right\|^2 + \left\|\bar{\theta}^{(t)}\right\|^2 - 2(\bar{\theta}^{(t)})^\top\gamma^{(t+1)}$$

$$= \operatorname{Tr}\left(\operatorname{Cov}(\bar{\theta}^{(t+1)})\right) + \left\|\mathbb{E}_{\xi^{(t)}}[\bar{\theta}^{(t+1)}-\bar{\theta}^{(t)}]\right\|^2$$

$$= \operatorname{Tr}\left(\operatorname{Cov}(\frac{\eta}{m}\sum_{k=1}^m \nabla\mathcal{L}(\theta_k^{(t)};\xi_k^{(t)}))\right) + \left\|\mathbb{E}_{\xi^{(t)}}[\bar{\theta}^{(t+1)}-\bar{\theta}^{(t)}]\right\|^2, \tag{D.4}$$

where the second equality follows from the definition of the covariance matrix, namely

$$\operatorname{Tr}\left(\operatorname{Cov}(\bar{\theta}^{(t+1)})\right) = \mathbb{E}_{\xi^{(t)}}\left\|\bar{\theta}^{(t+1)}\right\|^2 - \left\|\mathbb{E}_{\xi^{(t)}}\bar{\theta}^{(t+1)}\right\|^2,$$

and the final equality is derived from the original update of decentralized SGD (without applying Proposition D.3):

$$\bar{\theta}^{(t+1)} = \bar{\theta}^{(t)} - \eta \cdot \frac{1}{m}\sum_{k=1}^m \nabla\mathcal{L}(\theta_k^{(t)};\xi_k^{(t)}).$$

By convexity of the vector norm and the fact that

$$\operatorname{Tr}\left(\operatorname{Cov}(\frac{1}{m}\sum_{k=1}^m \nabla\mathcal{L}(\theta_k^{(t)};\xi_k^{(t)}))\right) = \mathbb{E}_{\xi^{(t)}}\left\|\frac{1}{m}\sum_{k=1}^m \nabla\mathcal{L}(\theta_k^{(t)};\xi_k^{(t)}) - \frac{1}{m}\sum_{k=1}^m \mathbb{E}_{\xi_k^{(t)}}\nabla\mathcal{L}(\theta_k^{(t)};\xi_k^{(t)})\right\|^2,$$

$$\tag{D.5}$$

we then complete the proof by applying Proposition D.3 and the bounded noise assumption in Assumption 3. $\square$

**Corollary D.5.** *Let* $\Gamma^{(t)} = \frac{1}{m}\sum_{k=1}^m (\theta_k^{(t)} - \bar{\theta}^{(t)})(\theta_k^{(t)} - \bar{\theta}^{(t)})^\top \in \mathbb{R}^{d\times d}$, *where* $\bar{\theta}^{(t)} = \frac{1}{m}\sum_{k=1}^m \theta_k^{(t)} \in \mathbb{R}^d$ *denotes the globally averaged model across* $m$ *agents. Suppose Assumption 2 holds. Then, for* $\epsilon^{(t)} \sim \mathcal{N}(0, \Gamma^{(t)})$, *the expected gradient perturbation satisfies:*

$$\mathbb{E}_{\epsilon^{(t)}\sim\mathcal{N}(0,\Gamma^{(t)})}\left[\nabla\mathcal{L}(\bar{\theta}^{(t)} + \epsilon^{(t)})\right] - \nabla\mathcal{L}(\bar{\theta}^{(t)})$$
$$= \frac{1}{2}\nabla\operatorname{Tr}\left(\nabla^2\mathcal{L}(\bar{\theta}^{(t)})\Gamma^{(t)}\right) + \mathbb{E}_{\epsilon^{(t)}\sim\mathcal{N}(0,\Gamma^{(t)})}\left[R_3(\epsilon^{(t)})\right], \qquad \text{(D.6)}$$

*where* $\|R_3(\epsilon^{(t)})\|$ *is bounded by* $\frac{L_4}{24}\|\epsilon^{(t)}\|^3$.

*Proof.* We apply the third-order Taylor expansion to $\nabla\mathcal{L}$ around $\bar{\theta}^{(t)}$:

$$\nabla\mathcal{L}(\bar{\theta}^{(t)} + \epsilon^{(t)}) = \nabla\mathcal{L}(\bar{\theta}^{(t)}) + \nabla^2\mathcal{L}(\bar{\theta}^{(t)})\epsilon^{(t)} + \frac{1}{2}\nabla^3\mathcal{L}(\bar{\theta}^{(t)})[\epsilon^{(t)}, \epsilon^{(t)}] + R_3(\epsilon^{(t)}),$$

with the remainder:

$$R_3(\epsilon^{(t)}) = \int_0^1 \frac{(1-\tau)^3}{6}\nabla^4\mathcal{L}(\bar{\theta}^{(t)} + \tau\epsilon^{(t)})[\epsilon^{(t)}, \epsilon^{(t)}, \epsilon^{(t)}]d\tau.$$

Taking expectations over $\epsilon^{(t)} \sim \mathcal{N}(0, \Gamma^{(t)})$, since $\mathbb{E}[\epsilon^{(t)}] = 0$, the linear term vanishes. The quadratic term $\mathbb{E}\left[\nabla^3\mathcal{L}(\bar{\theta}^{(t)})[\epsilon^{(t)}, \epsilon^{(t)}]\right]$ simplifies to $\nabla\operatorname{Tr}\left(\nabla^2\mathcal{L}(\bar{\theta}^{(t)})\Gamma^{(t)}\right)$ due to properties of the Gaussian distribution. The remainder can be bounded as

$$\|R_3(\epsilon^{(t)})\| \leq \int_0^1 \frac{(1-\tau)^3}{6}L_4\|\epsilon^{(t)}\|^3 d\tau = L_4\|\epsilon^{(t)}\|^3 \cdot \frac{1}{6}\int_0^1 (1-\tau)^3 d\tau.$$

Since $\int_0^1 (1-\tau)^3 d\tau = \frac{1}{4}$, we have:

$$\|R_3(\epsilon^{(t)})\| \leq L_4\|\epsilon^{(t)}\|^3 \cdot \frac{1}{6} \cdot \frac{1}{4} = \frac{L_4}{24}\|\epsilon^{(t)}\|^3.$$

$\square$

For comparison, we restate the convergence rate of DSGD from Koloskova et al. (2020).

**Assumption D.1** (*L*-smoothness). *Each population risk* $\mathcal{L}_k = \mathbb{E}_{\xi_k\sim\mathcal{D}_k}\mathcal{L}(\theta;\xi_k)$ *for* $k \in \{1, \ldots, m\}$ *is continuously differentiable, and there is a constant* $L \geq 0$ *such that:*

$$\|\nabla\mathcal{L}_k(\theta) - \nabla\mathcal{L}_k(\vartheta)\| \leq L\|\theta - \vartheta\|, \quad \forall\theta, \vartheta \in \mathbb{R}^d. \qquad \text{(D.7)}$$

**Theorem D.6** (Non-convex Convergence Rate of DSGD (Koloskova et al., 2020)). *Under Assumption 1, Assumption D.1 and Assumption 3, let the learning rate* $\eta$ *satisfy* $\eta \leq \eta_{\max} = \mathcal{O}\left(\frac{p}{L}\right)$, *and let* $\bar{\theta}^{(t)} = \frac{1}{m}\sum_{k=1}^m \theta_k^{(t)}$ *denote the averaged model at the* $t$-*th step. To achieve an* $\varepsilon$-*stationary point such that* $\frac{1}{T}\sum_{t=0}^{T-1}\mathbb{E}\left[\|\nabla\mathcal{L}(\bar{\theta}^{(t)})\|_2^2\right] \leq \varepsilon$, *the total number of steps* $T$ *satisfies:*

$$T = \mathcal{O}\left(\frac{\sigma^2}{m\,\varepsilon^2} + \frac{\sqrt{p}\,\sigma + \zeta}{p\,\varepsilon^{3/2}} + \frac{1}{p\varepsilon}\right) \cdot \left(\mathcal{L}(\theta_0) - \mathcal{L}^\star\right).$$

We then provide our main theoretical results as follows.

**Theorem D.7** (Non-convex Convergence Rate of DSGD). *Suppose Assumption 2 and Assumption 3 hold. Consider decentralized SGD (DSGD) with initializations* $\theta_k^{(0)} = \theta^{(0)}$ *for all* $k \in \mathcal{V}$, *and a constant learning rate satisfying* $\eta < \frac{2}{L_2}$. *Let* $\bar{\theta}^{(t)} = \frac{1}{m}\sum_{k=1}^m \theta_k^{(t)}$ *denote the averaged model at the* $t$-*th step. To achieve an* $\varepsilon$-*stationary point such that* $\frac{1}{T}\sum_{t=0}^{T-1}\mathbb{E}\left[\|\nabla\mathcal{L}(\bar{\theta}^{(t)})\|_2^2\right] \leq \varepsilon$, *the total number of steps* $T$ *satisfies:*

$$T = \mathcal{O}\left(\frac{\sigma^2}{m\varepsilon^2} + \frac{1}{\varepsilon} + \boxed{\sum_{t=0}^{T-1} U^{(t)}}\right) \cdot \left(\mathcal{L}(\theta^{(0)}) - \mathcal{L}^\star\right),$$

*where* $U^{(t)} = \frac{1}{2}(\eta L_2 - 1)\nabla\mathcal{L}(\bar{\theta}^{(t)})^\top\nabla\operatorname{Tr}\left(\nabla^2\mathcal{L}(\bar{\theta}^{(t)})\Gamma^{(t)}\right) + \Theta(\Xi_t^3)$, *with* $\Gamma^{(t)} = \frac{1}{m}\sum_{k=1}^m (\theta_k^{(t)} - \bar{\theta}^{(t)})(\theta_k^{(t)} - \bar{\theta}^{(t)})^\top$ *and the consensus distance* $\Xi_t^2 = \operatorname{Tr}(\Gamma^{(t)})$.

*Proof.* We structure the proof into several key steps.

**Step (A): Descent Force Decomposition.**

By $L_2$-smoothness (Assumption D.1) of the loss function $\mathcal{L}$ (as implied by Assumption 2), we can apply the first-order Taylor expansion around $\bar{\theta}^{(t)}$ to establish an upper bound for $\mathcal{L}(\bar{\theta}^{(t+1)})$:

$$\mathcal{L}(\bar{\theta}^{(t+1)}) \leq \mathcal{L}(\bar{\theta}^{(t)}) + \nabla\mathcal{L}(\bar{\theta}^{(t)})^\top (\bar{\theta}^{(t+1)} - \bar{\theta}^{(t)}) + \frac{L_2}{2}\|\bar{\theta}^{(t+1)} - \bar{\theta}^{(t)}\|^2.$$

According to Proposition D.3, we have

$$\mathbb{E}_{\xi^{(t)}}[\bar{\theta}^{(t+1)}] = \bar{\theta}^{(t)} - \eta\left(\nabla\mathcal{L}(\bar{\theta}^{(t)}) + \frac{1}{2}\nabla\operatorname{Tr}(\nabla^2\mathcal{L}(\bar{\theta}^{(t)})\Gamma^{(t)})\right) + \delta^{(t)},$$

where $\Gamma^{(t)}$ denotes the variance matrix of $\epsilon^{(t)} \sim \mathcal{N}(0, \Gamma^{(t)})$ and $\delta^{(t)} = \Theta\left(\frac{\eta}{m}\sum_{k=1}^m \|\theta_k^{(t)} - \bar{\theta}^{(t)}\|_2^3\right)$ denotes the higher-order residuals (see Proposition D.3).

Substituting this into the previous bound and taking the expectation with respect to random data sampling yields:

$$\mathbb{E}_{\xi^{(t)}}[\mathcal{L}(\bar{\theta}^{(t+1)})]$$
$$\leq \mathcal{L}(\bar{\theta}^{(t)}) - \eta\nabla\mathcal{L}(\bar{\theta}^{(t)})^\top\left(\nabla\mathcal{L}(\bar{\theta}^{(t)}) + \frac{1}{2}\nabla\operatorname{Tr}(\nabla^2\mathcal{L}(\bar{\theta}^{(t)})\Gamma^{(t)}) - \delta^{(t)}\right) + \mathbb{E}_{\xi^{(t)}}\frac{L_2}{2}\|\bar{\theta}^{(t+1)} - \bar{\theta}^{(t)}\|^2.$$

According to Corollary D.4, we obtain

$$\mathbb{E}_{\xi^{(t)}}\|\bar{\theta}^{(t+1)} - \bar{\theta}^{(t)}\|^2 \leq \frac{\eta^2\sigma^2}{m} + \eta^2\|\nabla\mathcal{L}(\bar{\theta}^{(t)}) + \frac{1}{2}\nabla\operatorname{Tr}(\nabla^2\mathcal{L}(\bar{\theta}^{(t)})\Gamma^{(t)}) + \delta^{(t)}\|^2.$$

We can then decompose the squared norm:

$$\left\|\nabla\mathcal{L}(\bar{\theta}^{(t)}) + \frac{1}{2}\nabla\operatorname{Tr}(\nabla^2\mathcal{L}(\bar{\theta}^{(t)})\Gamma^{(t)})\right\|^2$$
$$= \frac{1}{4}\left\|\nabla\operatorname{Tr}(\nabla^2\mathcal{L}(\bar{\theta}^{(t)})\Gamma^{(t)})\right\|^2 + \left\|\nabla\mathcal{L}(\bar{\theta}^{(t)})\right\|^2 + \nabla\operatorname{Tr}(\nabla^2\mathcal{L}(\bar{\theta}^{(t)})\Gamma^{(t)})^\top\mathcal{L}(\bar{\theta}^{(t)}).$$

Combining the previous steps, we obtain:

$$\mathbb{E}_{\xi^{(t)}}\mathcal{L}(\bar{\theta}^{(t+1)}) \leq \mathcal{L}(\bar{\theta}^{(t)}) - (\eta - \frac{\eta^2 L_2}{2})\left\|\nabla\mathcal{L}(\bar{\theta}^{(t)})\right\|^2 + \frac{\eta^2 L_2}{8}\underbrace{\left\|\nabla\operatorname{Tr}(\nabla^2\mathcal{L}(\bar{\theta}^{(t)})\Gamma^{(t)})\right\|^2}_{T_1}$$

$$\frac{1}{2}(-\eta + \eta^2 L_2)\cdot\underbrace{\nabla\mathcal{L}(\bar{\theta}^{(t)})^\top\nabla\operatorname{Tr}(\nabla^2\mathcal{L}(\bar{\theta}^{(t)})\Gamma^{(t)})}_{T_2} + \eta\underbrace{\nabla\mathcal{L}(\bar{\theta}^{(t)})^\top\delta^{(t)}}_{T_3} + \frac{\eta^2\sigma^2}{m}\cdot\frac{L_2}{2}$$

$$+ \eta^2 L_2\underbrace{\left(\nabla\mathcal{L}(\bar{\theta}^{(t)}) + \frac{1}{2}\nabla\operatorname{Tr}(\nabla^2\mathcal{L}(\bar{\theta}^{(t)})\Gamma^{(t)})\right)^\top\delta^{(t)}}_{T_4} + \frac{\eta^2 L_2}{2}\underbrace{\left\|\delta^{(t)}\right\|^2}_{T_5}. \qquad \text{(D.8)}$$

We subsequently control terms related to $\mathbb{E}_{\epsilon^{(t)} \sim \mathcal{N}(0, \Gamma^{(t)})}\nabla\mathcal{L}(\bar{\theta}^{(t+\frac{1}{2})}) - \nabla\mathcal{L}(\bar{\theta}^{(t)})$ in Equation (D.8).

**Step (B): Control Consensus-Related Terms**

Combining the residual upper bound in Corollary D.5 with the concavity of $(\cdot)^{3/2}$, we can derive

$$\left\|\delta^{(t)}\right\| \leq \frac{L_4}{24}\cdot\frac{1}{m}\sum_{k=1}^m\left\|\theta_k^{(t)} - \bar{\theta}^{(t)}\right\|^3 \leq \frac{L_4}{24}\cdot\sqrt{m}\left(\frac{1}{m}\sum_{k=1}^m\left\|\theta_k^{(t)} - \bar{\theta}^{(t)}\right\|^2\right)^{\frac{3}{2}},$$

and thus $T_3 \leq \frac{L_1 L_4}{24}\cdot\sqrt{m}\left(\frac{1}{m}\sum_{k=1}^m\left\|\theta_k^{(t)} - \bar{\theta}^{(t)}\right\|^2\right)^{\frac{3}{2}}$.

Further, by convexity of the squared norm, we have

$$T_5 = \left\|\delta^{(t)}\right\|^2 \leq \frac{m L_4^2}{24^2}\cdot\left(\frac{1}{m}\sum_{k=1}^m\left\|\theta_k^{(t)} - \bar{\theta}^{(t)}\right\|^2\right)^3.$$

According to Assumption 2, we can upper-bound $T_1$ as

$$T_1 = \left\| \nabla \operatorname{Tr}(\nabla^2 \mathcal{L}(\bar{\theta}^{(t)}) \Gamma^{(t)}) \right\|^2 = \left\| \nabla^3 \mathcal{L}(\bar{\theta}^{(t)}) : \Gamma^{(t)} \right\|^2 \leq L_3^2 \cdot \left( \frac{1}{m} \sum_{k=1}^{m} \left\| \theta_k^{(t)} - \bar{\theta}^{(t)} \right\|^2 \right)^2,$$

where the colon (:) represents the tensor contraction of the third-order derivative with the matrix $\Gamma$.

We can also bound $T_4$ as follows:

$$T_4 \leq \left\| \frac{1}{2} \nabla \operatorname{Tr}(\nabla^2 \mathcal{L}(\bar{\theta}^{(t)}) \Gamma^{(t)}) + \nabla \mathcal{L}(\bar{\theta}^{(t)}) \right\| \cdot \frac{1}{m} \sum_{k=1}^{m} \left\| \theta_k^{(t)} - \bar{\theta}^{(t)} \right\|^3$$

$$\leq \left( \frac{L_3}{2} \frac{1}{m} \sum_{k=1}^{m} \left\| \theta_k^{(t)} - \bar{\theta}^{(t)} \right\|^2 + L_1 \right) \frac{1}{m} \sum_{k=1}^{m} \left\| \theta_k^{(t)} - \bar{\theta}^{(t)} \right\|^3$$

$$\leq \left( \frac{L_3}{2} \frac{1}{m} \sum_{k=1}^{m} \left\| \theta_k^{(t)} - \bar{\theta}^{(t)} \right\|^2 + L_1 \right) \sqrt{m} \left( \frac{1}{m} \sum_{k=1}^{m} \left\| \theta_k^{(t)} - \bar{\theta}^{(t)} \right\|^2 \right)^{\frac{3}{2}}.$$

For clarity, we consolidate the terms involving $T_1$ through $T_5$ in Equation (D.8):

$$\text{Accumulated } T\text{-terms} = (-\eta + \eta^2 L_2) T_2 + \eta T_3 + \frac{\eta^2 L_2}{8} (T_1 + 8T_4 + 4T_5). \tag{D.9}$$

Substituting the Upper bounds for the Accumulated $T$-terms into Equation (D.8) yields the updated descent inequality:

$$\mathbb{E}_{\xi^{(t)}} \mathcal{L}(\bar{\theta}^{(t+1)}) \leq \mathcal{L}(\bar{\theta}^{(t)}) - \left( \eta - \frac{\eta^2 L_2}{2} \right) \left\| \nabla \mathcal{L}(\bar{\theta}^{(t)}) \right\|^2 + \eta U^{(t)} + \frac{\eta^2 \sigma^2 L_2}{2}, \tag{D.10}$$

where $\eta U^{(t)}$ denotes the upper bound for Equation (D.9):

$$\eta U^{(t)} \triangleq (\eta^2 L_2 - \eta) T_2 + \frac{\eta \sqrt{m} L_1 L_4}{24} \Xi_t^3$$

$$+ \frac{1}{8} \eta^2 L_2 L_3^2 \Xi_t^4 + \frac{1}{2} \eta^2 \sqrt{m} L_2 (2L_1 + L_3 \Xi_t^2) \Xi_t^3 + \frac{\eta^2 m L_2 L_4^2}{1152} \Xi_t^6,$$

$$= \underbrace{(\eta^2 L_2 - \eta) T_2}_{\triangleq A^{(t)} = \Theta(\Xi_t^2)} + \underbrace{\left[ \left( \eta^2 L_2 + \frac{\eta L_4}{24} \right) \sqrt{m} L_1 + \frac{\eta^2}{8} L_2 L_3^2 \Xi_t + \frac{\eta^2}{2} \sqrt{m} L_2 L_3 \Xi_t^2 + \frac{\eta^2 m L_2 L_4^2}{1152} \Xi_t^3 \right] \Xi_t^3}_{\triangleq H^{(t)} = O(\Xi_t^3)}. \tag{D.11}$$

Recall that the consensus distance $\Xi_t^2 = \frac{1}{m} \sum_{k=1}^{m} \left\| \theta_k^{(t)} - \bar{\theta}^{(t)} \right\|^2$. To facilitate subsequent analysis, we further separate $U^{(t)}$ into an Acceleration term $A^{(t)}$ plus High-order terms $H^{(t)}$.

With $U^{(t)}$ serving as a unified proxy for the consensus errors, we derive the new rate as follows.

### Step (C): Derive the Convergence Rate

Starting from the descent inequality Equation (D.10):

$$\mathbb{E}_{\xi^{(t)}} \left[ \mathcal{L}(\bar{\theta}^{(t+1)}) \right] \leq \mathcal{L}(\bar{\theta}^{(t)}) - \left( \eta - \frac{\eta^2 L_2}{2} \right) \left\| \nabla \mathcal{L}(\bar{\theta}^{(t)}) \right\|^2 + \eta U^{(t)} + \frac{\sigma^2}{m} \frac{\eta^2 L_2}{2}.$$

Taking full expectation and summing over $t = 0, \ldots, T-1$, we obtain

$$\sum_{t=0}^{T-1} \left( \eta - \frac{\eta^2 L_2}{2} \right) \mathbb{E} \left\| \nabla \mathcal{L}(\bar{\theta}^{(t)}) \right\|^2 \leq \mathcal{L}(\theta^{(0)}) - \mathbb{E} \left[ \mathcal{L}(\bar{\theta}^{(T)}) \right] + \eta \sum_{t=0}^{T-1} U^{(t)} + \frac{\sigma^2 \eta^2 L_2 T}{2m}.$$

To ensure the descent property of $-\left\| \nabla \mathcal{L}(\bar{\theta}^{(t)}) \right\|^2$, we require $\eta - \frac{\eta^2 L_2}{2} \geq 0$, which in turn implies that $\eta \leq \frac{\ell}{L_2}$, with $\ell < 2$. Under this condition, and denoting $\Delta = \mathcal{L}(\bar{\theta}^{(0)}) - \mathcal{L}^*$, we obtain

$$\frac{1}{T} \sum_{t=0}^{T-1} \mathbb{E} \left\| \nabla \mathcal{L}(\bar{\theta}^{(t)}) \right\|^2 \leq \frac{2\Delta}{(2-\ell) \eta T} + \frac{2}{(2-\ell) \eta} \frac{1}{T} \sum_{t=0}^{T-1} U^{(t)} + \frac{\sigma^2 \eta L_2}{(2-\ell) m}. \tag{D.12}$$

To ensure this is at most $\varepsilon$, it suffices to enforce

$$\frac{\sigma^2 \eta L_2}{(2-\ell)\,m} \le \frac{\varepsilon}{3}, \quad \frac{2}{(2-\ell)}\frac{1}{T}\sum_{t=0}^{T-1} U^{(t)} \le \frac{\varepsilon}{3}, \quad \text{and} \quad \frac{2\Delta}{(2-\ell)\,\eta\,T} \le \frac{\varepsilon}{3}. \tag{D.13}$$

To satisfy all three conditions simultaneously, along with a stability condition $\eta \le \frac{\ell}{L_2}$, we should select $\eta$ accordingly:

$$\eta \;\le\; \min\left\{\frac{\ell}{L_2},\; \frac{(2-\ell)m\varepsilon}{3\sigma^2 L_2}\right\}, \quad \text{and} \quad T \ge \max\left\{\frac{6\,\Delta}{(2-\ell)\eta\varepsilon},\; \frac{6}{(2-\ell)\varepsilon}\sum_{t=0}^{T-1} U^{(t)}\right\}.$$

To ensure a valid step-size $\eta$ exists, we substitute these three upper bounds into the condition for $T$. This yields three distinct lower bounds on the total number of iterations $T$ that must be satisfied. By rearranging the inequality $T\eta \ge \frac{6\Delta}{\varepsilon}$, we require:

$$T \;\ge\; \max\left\{\frac{6\Delta L_2}{\ell\,(2-\ell)\,\varepsilon},\; \frac{18\Delta\sigma^2 L_2}{(2-\ell)^2\,m\varepsilon^2},\; \frac{6}{(2-\ell)\varepsilon}\sum_{t=0}^{T-1} U^{(t)}\right\},$$

where the first two bounds are derived directly by substituting the first two terms from the $\min\{\cdot\}$ operation for $\eta$ into the first lower bound of $T$.

Therefore, the total number of iterations $T$ should be large enough to satisfy all applicable lower bounds. This leads to the sufficient condition:

$$T = \mathcal{O}\left(\frac{\Delta}{\varepsilon} + \frac{\Delta\,\sigma^2}{m\,\varepsilon^2} + \frac{1}{\varepsilon}\sum_{t=0}^{T-1} U^{(t)}\right),$$

This condition is sufficient to guarantee

$$\frac{1}{T}\sum_{t=0}^{T-1} \mathbb{E}\big\|\nabla\mathcal{L}(\bar{\theta}^{(t)})\big\|_2^2 \le \varepsilon.$$

The proof is now complete. $\qquad\qquad\square$

---

**Proposition D.8.** *Suppose Assumption 2 and Assumption 4 hold. Assume $\eta > 1/L_2$, and assume $\|\nabla\mathcal{L}(\bar{\theta}^{(t)})\| \ge \mu_t > 0$ for all $t$. Consider the matrix $\Gamma^{(t)} = \frac{1}{m}\sum_{k=1}^{m}(\theta_k^{(t)} - \bar{\theta}^{(t)})(\theta_k^{(t)} - \bar{\theta}^{(t)})^\top$ and its trace $\Xi_t^2 = \mathrm{Tr}(\Gamma^{(t)})$. Then, for any fixed $m > 0$, there exists $\Xi_t^2 > 0$ such that*

$$U^{(t)} \triangleq \frac{1}{2}(\eta L_2 - 1)\underbrace{\nabla\mathcal{L}(\bar{\theta}^{(t)})^\top\nabla\,\mathrm{Tr}\big(\nabla^2\mathcal{L}(\bar{\theta}^{(t)})\,\Gamma^{(t)}\big)}_{=\Theta(\Xi_t^2)} + \Theta(\Xi_t^3) < 0. \tag{D.14}$$

---

**Remark D.1.** We note that Proposition D.8 does not contradict Equation (D.12) when both $\Delta$ and $\sigma$ are zero. The condition $\Delta = \mathcal{L}(\bar{\theta}^{(0)}) - \mathcal{L}^* = 0$ implies that the models are initialized at an optimal point. In Theorem D.7, we assume that all initializations are identical ($\theta_k^{(0)} = \theta^{(0)}, \forall k \in \mathcal{V}$), so it follows that all models begin at the same optimum. Consequently, the consensus error remains zero throughout all iterations, meaning the model covariance matrix $\Gamma^{(t)}$ is the zero matrix and its trace $\Xi_t$ is also zero. Since every component of the term $U^{(t)}$, defined as

$$U^{(t)} \triangleq \underbrace{(\eta L_2 - 1)T_2}_{\triangleq A^{(t)} = \Theta(\Xi_t^2)} + \underbrace{\left[\left(\eta L_2 + \frac{L_4}{24}\right)\sqrt{m}L_1 + \frac{1}{8}\eta L_2 L_3^2\Xi_t + \frac{1}{2}\eta\sqrt{m}L_2 L_3\Xi_t^2 + \frac{\eta m L_2 L_4^2}{1152}\Xi_t^3\right]\Xi_t^3}_{\triangleq H^{(t)} = O(\Xi_t^3)}. \tag{D.15}$$

*Proof.* The proof relies on establishing that for sufficiently small $\Xi_t$, the negative leading term $A^{(t)}$ in the decomposition of $U^{(t)}$ dominates the higher-order residual term $H^{(t)}$. Specifically, $A^{(t)}$ is of order $\Theta(\Xi_t^2)$, while $H^{(t)}$ is of order $O(\Xi_t^3)$.

**Step (A): Derive Upper Bound on $A^{(t)}$.** Let $g^{(t)} = \nabla \mathcal{L}(\bar{\theta}^{(t)})$ and recall that

$$T_2 \triangleq (g^{(t)})^\top \nabla \operatorname{Tr}\big(\nabla^2 \mathcal{L}(\bar{\theta}^{(t)})\, \Gamma^{(t)}\big).$$

Based on the definition of $U^{(t)}$ in Equation (D.15), we have the leading term $A^{(t)} = (\eta L_2 - 1)T_2$. To analyze $T_2$, consider the trilinear form defined on the unit sphere. Let $\delta_k^{(t)}$ be the deviation vectors such that $\Gamma^{(t)} = \frac{1}{m} \sum_{k=1}^m \delta_k^{(t)}(\delta_k^{(t)})^\top$ and $\Xi_t^2 = \frac{1}{m} \sum_{k=1}^m \|\delta_k^{(t)}\|^2$. Define the function $F(\delta)$ for $\delta \neq 0$:

$$F(\delta) = \frac{\nabla^3 \mathcal{L}(\bar{\theta}^{(t)})[\delta, \delta, g^{(t)}]}{\|g^{(t)}\|\|\delta\|^2}.$$

Under Assumption 4, we have $\nabla^3 \mathcal{L}(\bar{\theta}^{(t)})[\delta, \delta, g^{(t)}] < 0$, implying $F(\delta) < 0$. Since the unit sphere $S = \{\delta : \|\delta\| = 1\}$ is compact, $F$ attains a maximum value $M < 0$. Let $\gamma = -M > 0$. It follows that for any vector $\delta$,

$$\nabla^3 \mathcal{L}(\bar{\theta}^{(t)})[\delta, \delta, g^{(t)}] \leq -\gamma \|g^{(t)}\|\|\delta\|^2.$$

Substituting this bound into the summation for $T_2$:

$$T_2 = \frac{1}{m} \sum_{k=1}^m \nabla^3 \mathcal{L}(\bar{\theta}^{(t)})[\delta_k^{(t)}, \delta_k^{(t)}, g^{(t)}] \leq \frac{1}{m} \sum_{k=1}^m \big(-\gamma \|g^{(t)}\|\|\delta_k^{(t)}\|^2\big) = -\gamma \|g^{(t)}\|\Xi_t^2.$$

Given the gradient lower bound $\|g^{(t)}\| \geq \mu_t$, we obtain $T_2 \leq -\gamma \mu_t \Xi_t^2$. Consequently, assuming the pre-condition $\eta L_2 - 1 > 0$ holds, the leading term $A^{(t)}$ satisfies:

$$A^{(t)} = (\eta L_2 - 1)T_2 \leq -(\eta L_2 - 1)\gamma \mu_t \Xi_t^2. \tag{D.16}$$

This confirms that $A^{(t)}$ provides a strictly negative contribution of order $\Theta(\Xi_t^2)$.

**Step (B): Dominance over Higher-Order Residuals.** We now show that $U^{(t)} = A^{(t)} + H^{(t)} < 0$ for small $\Xi_t$. According to Equation (D.15), the residual term is given by

$$H^{(t)} = \underbrace{\left[\left(\eta L_2 + \frac{L_4}{24}\right)\sqrt{m}L_1 + \frac{1}{8}\eta L_2 L_3^2 \Xi_t + \frac{1}{2}\eta\sqrt{m}L_2 L_3 \Xi_t^2 + \frac{\eta m L_2 L_4^2}{1152}\Xi_t^3\right]}_{\triangleq P(\Xi_t)} \Xi_t^3.$$

Here, $P(\Xi_t)$ is a polynomial in $\Xi_t$ with positive coefficients, and thus $H^{(t)} = O(\Xi_t^3)$. The condition $U^{(t)} < 0$ is equivalent to $H^{(t)} < -A^{(t)}$. Using the bound from Equation (D.16), it suffices to show:

$$P(\Xi_t)\Xi_t^3 < (\eta L_2 - 1)\gamma \mu_t \Xi_t^2.$$

For $\Xi_t > 0$, we divide both sides by $\Xi_t^2$, reducing the condition to:

$$\Xi_t \cdot P(\Xi_t) < (\eta L_2 - 1)\gamma \mu_t.$$

Let $Q(u) = u \cdot P(u)$ for $u \geq 0$. Since $P(u)$ is bounded in a neighborhood of zero, we have:

$$\lim_{u \to 0^+} Q(u) = 0 \cdot \left[\left(\eta L_2 + \frac{L_4}{24}\right)\sqrt{m}L_1\right] = 0.$$

The term on the right-hand side, $C \triangleq (\eta L_2 - 1)\gamma \mu_t$, is a strictly positive constant. By the continuity of polynomial functions and the intermediate value theorem, there exists a threshold $\delta > 0$ such that for all $0 < \Xi_t < \delta$, the inequality $Q(\Xi_t) < C$ holds. This implies that for a sufficiently small consensus error $\Xi_t$, the negative drift from $A^{(t)}$ dominates the residual $H^{(t)}$, ensuring $U^{(t)} < 0$. $\quad\square$

**Explanation.** The high-level intuition of Theorem D.7 and Proposition D.8 is outlined by the following descent lemma.

$$\mathbb{E}_{\xi^{(t)}}\left[\mathcal{L}(\bar{\theta}^{(t+1)})\right] \leq \mathcal{L}(\bar{\theta}^{(t)}) - \underbrace{\left(\eta - \frac{\eta^2 L_2}{2}\right)}_{>0} \underbrace{\left\|\nabla \mathcal{L}(\bar{\theta}^{(t)})\right\|^2}_{\text{Standard Descent}}$$

$$+ \underbrace{\left(\eta^2 L_2 - \eta\right)}_{>0} \underbrace{\nabla \mathcal{L}(\bar{\theta}^{(t)})^\top \nabla \mathrm{Tr}\left(\nabla^2 \mathcal{L}(\bar{\theta}^{(t)}) \Gamma^{(t)}\right)}_{<0, \text{ progressive sharpening}} + \frac{\eta^2 L_2 \sigma^2}{2m} + \mathcal{O}(\Xi_t^3).$$

$$\text{(D.17)}$$

**Remark D.2.** We need to satisfy both the requirement $\frac{\sigma^2 \eta L_2}{(2-\ell)m} \leq \varepsilon/3$ in Equation (D.13) and the condition $\eta > 1/L_2$ needed to maintain the descent property introduced by progressive sharpening. We can satisfy both conditions by appropriately scaling the number of agents $m$: for any arbitrarily small target accuracy $\varepsilon$, $\frac{\sigma^2 \eta L_2}{(2-\ell)m} \leq \varepsilon/3$ is guaranteed to hold as long as the network size satisfies $m > \frac{3\sigma^2}{(2-\ell)\varepsilon}$. In other words, we can achieve the same convergence rate as parallel SGD by adapting the network size $m$ to the desired target accuracy $\varepsilon$.

**Proposition D.9** (Critical Consensus Edge). *Suppose Assumption 1-Assumption 3 hold. Assume $\eta > \frac{1}{L_2}$ and that the consensus error satisfies $\Xi_t \leq 1$ for all t. Then the following condition ensures that the critical condition in Inequality (8) is satisfied:*

$$\sqrt{\frac{24(1-p)\eta^2}{p^2}\left(\phi^2 + \sigma^2\right)} < \min\left\{ \frac{(\eta L_2 - 1)\gamma^* \mu_t}{2(\eta L_2 + \frac{L_4}{24})\sqrt{m}L_1}, \quad \sqrt{\frac{(\eta L_2 - 1)\gamma^* \mu_t}{2\Sigma_{\text{high}}}} \right\}, \quad \text{(D.18)}$$

*where $\Sigma_{\text{high}} = \frac{1}{8}\eta L_2 L_3^2 + \frac{1}{2}\eta\sqrt{m}L_2 L_3 + \frac{\eta m L_2 L_4^2}{1152}$. Here, $\gamma^*$ denotes the degree of progressive sharpening (see Assumption 4), $\phi^2$ denotes the uniform upper bound of the averaged squared local gradient norm (i.e., $\frac{1}{m}\sum_{k=1}^{m} \|\nabla \mathcal{L}_k(\theta_k^{(t)})\|^2 \leq \phi^2$), and $\mu_t$ is the lower bound on the global gradient norm (i.e., $\|\nabla \mathcal{L}(\bar{\theta}^{(t)})\| \geq \mu_t > 0$).*

*Proof.* The proof establishes that the condition in Equation (11) suffices to guarantee $U^{(t)} < 0$. Recall from the decomposition in Equation (D.15) that $U^{(t)} = A^{(t)} + H^{(t)}$, where $A^{(t)}$ is the leading descent term and $H^{(t)}$ represents higher-order residuals. We aim to show that the negative drift dominates the error, i.e., $H^{(t)} < -A^{(t)}$.

Using the lower bound on the gradient norm $\|\nabla \mathcal{L}(\bar{\theta}^{(t)})\| \geq \mu_t$ derived in Proposition 2, the leading term satisfies $A^{(t)} \leq -(\eta L_2 - 1)\gamma^* \mu_t \Xi_t^2$. Let $K \triangleq (\eta L_2 - 1)\gamma^* \mu_t$. Under the sharpening regime $(\eta L_2 > 1)$, we have $K > 0$. Thus, a sufficient condition for $U^{(t)} < 0$ is:

$$H^{(t)} < K\Xi_t^2. \quad \text{(D.19)}$$

**Step (A): Bounding the residual term.** Substituting the expansion of $H^{(t)}$ from Equation (D.15) into Equation (D.19) and dividing both sides by $\Xi_t^2$ (assuming $\Xi_t > 0$), the requirement becomes:

$$\underbrace{\left[\left(\eta L_2 + \frac{L_4}{24}\right)\sqrt{m}L_1 + \frac{1}{8}\eta L_2 L_3^2 \Xi_t + \frac{1}{2}\eta\sqrt{m}L_2 L_3 \Xi_t^2 + \frac{\eta m L_2 L_4^2}{1152}\Xi_t^3\right]}_{\triangleq P(\Xi_t)} \Xi_t < K.$$

We define $C_{\text{lin}} \triangleq (\eta L_2 + \frac{L_4}{24})\sqrt{m}L_1$ as the coefficient of the linear term. Using the assumption that the consensus error is locally bounded ($\Xi_t \leq 1$), we have $\Xi_t^k \leq \Xi_t$ for $k \geq 1$. This allows us to upper bound the higher-order polynomial terms using the aggregated coefficient $\Sigma_{\text{high}}$ defined in the proposition:

$$P(\Xi_t)\Xi_t \leq C_{\text{lin}}\Xi_t + \Sigma_{\text{high}}\Xi_t^2.$$

Therefore, it suffices to ensure $C_{\text{lin}}\Xi_t + \Sigma_{\text{high}}\Xi_t^2 < K$.

**Step (B).** To satisfy the inequality above, we employ a budget splitting strategy, requiring both the linear and quadratic components to be bounded by half of the descent budget $K/2$. This yields two separate constraints on $\Xi_t$:

$$C_{\text{lin}}\Xi_t < \frac{K}{2} \implies \Xi_t < \frac{K}{2C_{\text{lin}}}, \quad \text{and} \quad \Sigma_{\text{high}}\Xi_t^2 < \frac{K}{2} \implies \Xi_t < \sqrt{\frac{K}{2\Sigma_{\text{high}}}}.$$

Consequently, if $\Xi_t < \min\left\{\frac{K}{2C_{\mathrm{lin}}}, \sqrt{\frac{K}{2\Sigma_{\mathrm{high}}}}\right\}$, then $U^{(t)} < 0$ holds.

Finally, invoking Corollary D.2, which bounds the consensus error as $\Xi_t \leq \sqrt{\frac{24(1-p)\eta^2}{p^2}(\phi^2 + \sigma^2)}$, we see that Equation (11) ensures $\Xi_t$ falls within the safety region. This completes the proof. $\qquad\square$

