# OpenReview forum: "On The Surprising Effectiveness of a Single Global Merging in Decentralized Learning"
_ICLR.cc/2026/Conference — ICLR 2026 Oral_

### Official Review · Reviewer_jbK5 · 2025-10-26

**Soundness:** 3
**Presentation:** 3
**Contribution:** 3
**Rating:** 8
**Confidence:** 3

**Summary:**

The primary goal of the paper is to investigate optimal strategies for scheduling communication in decentralized learning, specifically focusing on when and how often devices should synchronize during training. The authors' empirical findings reveal that allocating most of the communication budget toward the end of training, particularly by performing a single, fully connected global merging at the final step, can achieve generalization performance comparable to traditional server-based (centralized) training, even under severe communication constraints and high data heterogeneity. On the theoretical side, the paper demonstrates that the globally merged model resulting from decentralized stochastic gradient descent (SGD) can converge as quickly as centralized mini-batch SGD, and provides new insights into why limited but well-timed communication is sufficient for effective model merging and generalization in decentralized settings

**Strengths:**

1. The paper tackles the core issues of communication bottlenecks and data heterogeneity in decentralized learning.
2. The paper investigates how to optimally schedule communication over time and links this to model merging.
3. The paper provides theoretical guarantees that the merged decentralized model can match or outperform centralized mini-batch SGD.
4. The paper shows that local models remain mergeable even with limited but nonzero communication.
5. Demonstrates findings with robust experiments on Tiny ImageNet, CIFAR-100 and CIFAR-10 datasets with up to 32 agents.

**Weaknesses:**

The decentralized setup in this paper assumes that all-reduce communication is impractical, so presenting a single global merging as an all-reduce operation may not be appropriate. Instead, a more suitable alternative is to perform simple gossip averaging (without additional SGD updates) for n iterations (where n is the number of nodes) when using a doubly stochastic graph structure. However, this shouldn't impact any of the results presented in the paper.

**Questions:**

Suggestions:
1. Can the authors add server-side training (centralized SGD) curve to the figure 3c? It would be insightful to add the centralized baseline there.
2. "Surprisingly, we uncover that fully connected communication at the final step. Low communication throughout training preserves the mergeability of local models." -- This phenomenon has already been shown in the literature [1] as Skew-Compensated Sparse Push in [1] but the current paper has no reference to [1]

References:
1. Aketi, S. A., Singh, A., & Rabaey, J. (2021). Sparse-push: Communication-& energy-efficient decentralized distributed learning over directed & time-varying graphs with non-iid datasets. arXiv preprint arXiv:2102.05715.

---

> ### Author Response · Authors · 2025-11-26
> **Response to Reviewer  jbK5 (Part 1/2)**
>
> We sincerely thank the reviewer for their strong support and for recognizing the significance of the core issues we addressed, alongside our theoretical guarantees and robust experiments. We address your remaining questions below.
>
> > **Q1**: The decentralized setup in this paper assumes that all-reduce communication is impractical, so presenting a single global merging as an all-reduce operation may not be appropriate. Instead, a more suitable alternative is to perform simple gossip averaging (without additional SGD updates) for n iterations (where n is the number of nodes) when using a doubly stochastic graph structure.
>
> **A1**: We thank the reviewer for this insightful comment regarding the practical realization of global merging. We agree that in decentralized settings, iterative gossip averaging is a more appropriate and practical implementation.
>
> Following your advice, we performed additional experiments using simple gossip averaging for multiple rounds after regular decentralized training. The results confirm that our findings are robust: this practical averaging strategy still leads to a significant improvement in test accuracy.
>
> Our new observations are as follows:
>
> - **Single Round of Final Gossip**: Surprisingly, a single round of gossip averaging already yields a significant performance jump, though it is slightly lower than a global merging.
>
> - **Multiple rounds of Final Gossip**: Only 5 rounds of gossip averaging (far fewer than the number of nodes $n$ as required) yield performance comparable to a global merging.
>
> The full results are provided in this **[link](https://anonymous.4open.science/r/Anonymous-Repo-for-Rebuttal-ICLR26-0E88/Additional%20Experiments/Final_Gossip_Merging.md)**.
>
> **Experiment Link:** https://anonymous.4open.science/r/Anonymous-Repo-for-Rebuttal-ICLR26-0E88/Additional%20Experiments/Final_Gossip_Merging.md

---

> > ### Author Response · Authors · 2025-11-26
> > **Response to Reviewer  jbK5 (Part 2/2)**
> >
> > > **Q2**: Can the authors add server-side training (centralized SGD) curve to the figure 3c? It would be insightful to add the centralized baseline there.
> >
> > **A2**: Thanks. We have revised Figure 3c to include the centralized baseline as requested. For your convenience, the updated figure is provided in the link below.
> >
> > **Experiment Link:** https://anonymous.4open.science/r/Anonymous-Repo-for-Rebuttal-ICLR26-0E88/Additional%20Experiments/Centralized_Baseline.md
> >
> > > **Q3**: "Surprisingly, we uncover that fully connected communication at the final step. Low communication throughout training preserves the mergeability of local models." -- This phenomenon has already been shown in the literature [1] as Skew-Compensated Sparse Push in [1] but the current paper has no reference to [1].
> >
> > **A3**: We thank the reviewer for highlighting this relevant work [1]. We have incorporated a detailed discussion of [1] in Section 4.2 of our revised manuscript.
> >
> > For your convenience, we provide the relevant discussion from our revision below:
> >
> > We note that [1] proposed Skew-Compensated Sparse Push (SCSP), an effective strategy to improve the communication efficiency of decentralized learning, which also includes a final global merging step. While both works share the goal of reducing communication, our approaches differ in  methodology and experimental setting: (1) *Methodology.* SCSP proposes a *gradient sparsification* algorithm (top-$k$ gradients) over a fixed topology. In contrast, we investigate the phenomenon of mergeability under *topological sparsification* (i.e., sparse gossip). (2) *Experimental setting.* Their analysis focuses on settings with a single local step ($H=1$). In contrast, we demonstrate that mergeability is remarkably robust even with a large number of local update steps (e.g., $H=100$) and high data heterogeneity.
> >
> > Below, we provide a more comprehensive comparison to further clarify the distinctions regarding our scientific scope, methodology, and geometric insights:
> >
> > **1. Scientific Scope**: Discovery vs. Proposal. Our core contribution is not the proposal of a final merge operation itself. Rather, it is the systematic discovery of the conditions under which local models remain "mergeable."
> >
> > - [1] primarily operates in settings with a single local step ($H=1$), where consensus is easier to maintain.
> > - In contrast, we demonstrate that the "mergeability" phenomenon is remarkably robust even with a large number of local update steps (e.g., $H=100$) and under severe data heterogeneity. This regime presents a far greater challenge, making the preservation of mergeability a surprising and novel finding.
> >
> > **2. Methodology**: The two papers employ distinct communication reduction mechanisms.
> > - [1] uses Top-$k$ sparsification, which filters the content of messages without restricting the graph topology.
> > - Our work investigates Sparse Gossip, where sparsity is defined by the network topology itself. These approaches address efficiency from different angles and are potentially complementary.
> >
> > **3. Geometric Insight**: Crucially, we provide the mechanistic explanation for why single merging works. Our analysis reveals that sparse gossip guides local models to align along a "high-loss ring" surrounding a central "low-loss basin" (Figure 1c). This geometric mechanism is unique to our work and is not analyzed in [1].
> >
> > ### Reference
> >
> > [1] Aketi, S. A., Singh, A., & Rabaey, J. (2021). Sparse-push: Communication-& energy-efficient decentralized distributed learning over directed & time-varying graphs with non-iid datasets. arXiv preprint arXiv:2102.05715.

---

### Official Review · Reviewer_pdQg · 2025-10-29

**Soundness:** 3
**Presentation:** 4
**Contribution:** 3
**Rating:** 8
**Confidence:** 4

**Summary:**

This paper studies how communication should be scheduled over time to improve global generalization. The authors show that a single global merging in decentralized training can achieve performance close to that of federated learning, and that limited but non-zero communication helps preserve the mergeability of local models throughout training.

In addition, the authors provide a theoretical explanation for why limited but non-zero communication can ensure mergeability, and why communication should be concentrated in the later stages of training. The investigated problem is novel, and the theoretical analysis appears sound. The findings may further inspire future research on communication scheduling and allocation.

**Strengths:**

This paper studies how communication should be scheduled over time to improve global generalization. The authors show that a single global merging in decentralized training can achieve performance close to that of federated learning, and that limited but non-zero communication helps preserve the mergeability of local models throughout training.

In addition, the authors provide a theoretical explanation for why limited but non-zero communication can ensure mergeability, and why communication should be concentrated in the later stages of training. The investigated problem is novel, and the theoretical analysis appears sound. The findings may further inspire future research on communication scheduling and allocation.

**Weaknesses:**

Comments:

1.	The paper does not discuss the impact of network topology in decentralized learning. In realistic decentralized settings, multiple agents are connected through an underlying topology, which may prevent a single global merging due to communication constraints. Under a general topology, if agents communicate randomly according to the topology and perform full communication within that topology—rather than global merging— in the final stage, would the main conclusions still hold?

2.	Regarding Proposition 3, could the authors provide more intuition on why communication is needed when the gradient norm is small? Intuitively, when the gradient norm is large, more communication should be beneficial to accelerate optimization; when the gradient norm is small—indicating that the model is approaching a stationary point—less communication should suffice. This seems to contradict the paper’s conclusion.

**Questions:**

N/A

---

> ### Author Response · Authors · 2025-11-26
> **Response to Reviewer pdQg**
>
> We sincerely thank the reviewer for their strong support and for recognizing the novelty of our investigated problem and the soundness of our theoretical analysis. We address your remaining questions below.
>
> > **Q1**: The paper does not discuss the impact of network topology in decentralized learning. In realistic decentralized settings, multiple agents are connected through an underlying topology, which may prevent a single global merging due to communication constraints. Under a general topology, if agents communicate randomly according to the topology and perform full communication within that topology—rather than global merging— in the final stage, would the main conclusions still hold?
>
> **A1**: We appreciate this constructive suggestion. It helps bridge the gap between our theoretical approach of global merging and the practical constraints of decentralized network topologies.
>
> The answer is yes. Following your advice, we performed additional experiments where agents communicate randomly according to a general topology and perform the final merge within that topology. The results confirm that our findings are robust: this topology-constrained final merging still leads to a significant improvement in test accuracy.
>
> Specifically, we observe that:
>
> - A single round of gossip merging still brings a significant performance jump, though it is slightly lower than the ideal global merging.
>
> -  Only 5 rounds of gossip without training, the performance gap between a global merging becomes marginal. The full results are provided in this **[link](https://anonymous.4open.science/r/Anonymous-Repo-for-Rebuttal-ICLR26-0E88/Additional%20Experiments/Final_Gossip_Merging.md)**.
>
> **Experiment Link:** https://anonymous.4open.science/r/Anonymous-Repo-for-Rebuttal-ICLR26-0E88/Additional%20Experiments/Final_Gossip_Merging.md
>
> > **Q2**: Regarding Proposition 3, could the authors provide more intuition on why communication is needed when the gradient norm is small? Intuitively, when the gradient norm is large, more communication should be beneficial to accelerate optimization; when the gradient norm is small—indicating that the model is approaching a stationary point—less communication should suffice. This seems to contradict the paper’s conclusion.
>
> **A2**: We clarify that there is no contradiction. The term $\|\nabla L(\theta)\|$ in Proposition 3 refers to the **gradient of the globally averaged model on global data** (we termed it as **“global” gradient** below), not the gradients of individual local models (**local gradients**). We agree with the reviewer that in an ideal IID scenario, communication might be less critical near convergence.
>
> However, in the heterogeneous (Non-IID) settings we consider, the dynamics are much different. The key distinction is that the **global gradient can be small (indicating the averaged model is in a good basin) while the local gradients remain large**. This is because, even when the global model reaches the desired global basin, local models are not necessarily converged. Driven by data heterogeneity, large local gradients point in different directions towards distinct local basins, constantly driving the models apart. We illustrate this phenomenon in this **[Figure](https://anonymous.4open.science/r/Anonymous-Repo-for-Rebuttal-ICLR26-0E88/Local%20and%20Global%20Forces%20in%20Decentralized%20Learning%20Loss%20Landscape.md)**.
>
>
> **Figure Link:** https://anonymous.4open.science/r/Anonymous-Repo-for-Rebuttal-ICLR26-0E88/Local%20and%20Global%20Forces%20in%20Decentralized%20Learning%20Loss%20Landscape.md
>
> **Intuition.** As illustrated in the the Figure, there is a **"tug-of-war"** between the Local Optimization Force (pulling towards local minima) and the Global Aggregation Force (pulling back via gossip). Proposition 3 suggests that we need more communication at the final stage to strengthen the Global Aggregation Force. This allows us to “counteract” strong local forces and maintain the averaged model in the optimal global basin.

---

> > ### Comment · Reviewer_pdQg · 2025-11-26
> >
> > Thanks to the authors for their efforts in preparing the response. It has addressed all my concerns.

---

> > > ### Author Response · Authors · 2025-11-26
> > > **Thank You for Your Continued Support**
> > >
> > > We are glad to hear that all your concerns have been addressed! Thank you for your continued support.

---

### Official Review · Reviewer_6EES · 2025-10-30

**Soundness:** 3
**Presentation:** 3
**Contribution:** 3
**Rating:** 6
**Confidence:** 4

**Summary:**

This paper showed that the performance of decentralized SGD can be improved by performing a single global merging at the end of the training. Then, this paper analyzed the convergence rate and provide the intuition of why the single global merging can improve the performance.

**Strengths:**

* It is an interesting finding that single global merging at the final training phase can significantly improve the accuracy of Decentralized SGD.
* This paper comprehensively evaluated how the effect of a single global merging affects the performance, depending on when we did it, showing that a single global merging is helpful at the final training phase.

**Weaknesses:**

* The experimental findings are interesting, but the theoretical explanation provided by this paper is difficult to follow.
* This paper claims that the convergence rate of Decentralized SGD with a single global merging matches the rate of Parallel SGD in Remark 2, but it is an overclaim. The convergence rate shown in Theorem 1 is not closed-form, which still depends on $\Xi_t$. Using Assumption 4, this term can disappear, but it is not fair to compare this rate and the convergence rate of Decentralized SGD since these analysis use the different assumptions.
* This paper claims that $A_t$ would be negative under Assumption 4 and the condition shown in Proposition 3 holds, but it is still unclear to the reviewer what Assumption 4 implies. Can the authors provide several functions that satisfy Assumption 4 or an intuition to understand Assumption 4?

Minor Comments:
* The statement of theorems and propositions must explicitly mention which assumptions are used. For instance, Theorem 1 and Propositions 2 and 3 seem to use Assumption 1, but it is not explicitly mentioned in their statements. Additionally, Proposition 2 seems to use Assumption 2, but it is not mentioned in the statement.
* This paper cited the same paper multiple times in the reference. (Kong 2021a and Kong 2021b)

**Questions:**

See the weakness section.

---

> ### Author Response · Authors · 2025-11-26
> **Response to Reviewer  6EES (Part 1/3)**
>
> We sincerely thank the reviewer for their positive assessment and for finding our discovery regarding the single global merging in decentralized learning comprehensive and interesting. We address your remaining questions on our theoretical analysis below.
>
> > **Q1**: This paper claims that the convergence rate of Decentralized SGD with a single global merging matches the rate of Parallel SGD in Remark 2, but it is an overclaim. The convergence rate shown in Theorem 1 is not closed-form, which still depends on $\Xi_t$
> . Using Assumption 4, this term can disappear, but it is not fair to compare this rate and the convergence rate of Decentralized SGD since these analysis use the different assumptions.
>
> **A1**: Thanks. We respectfully disagree and clarify that **our comparison is fair and holds under the same assumptions**. The confusion may stem from the behavior of the consensus error term in the baseline setting. For **Vanilla Parallel SGD**, local models are synchronized and identical at every step (i.e., $\theta\_k^{(t)} \equiv \bar{\theta}^{(t)}$ for all $k$).
>
> Consequently, the consensus error terms defined in our theorem become identically zero by definition:
> $$\Gamma^{(t)} \triangleq \frac{1}{m}\sum\_{k=1}^m (\theta\_k^{(t)} - \bar{\theta}^{(t)})(\theta\_k^{(t)} - \bar{\theta}^{(t)})^\top  \equiv 0, \quad \text{and} \quad $\Xi_t\triangleq \frac{1}{m}\sum\_{k=1}^m (\theta\_k^{(t)} - \bar{\theta}^{(t)})^\top(\theta\_k^{(t)} - \bar{\theta}^{(t)})\equiv 0$$
>
> Since these terms vanish naturally, the complex term $A_t \triangleq \eta L(2T_2 + L_3^2\Xi\_t^4 +(2L\_1 + 2L\_3\Xi\_t^2 + \frac{m L\_4^2}{24^2})\sqrt{m}\,\Xi\_t^3)$ where $T\_2 = (\nabla \mathcal{L}(\bar{\theta}^{(t)}))^\top \nabla \operatorname{Tr}(\nabla^2 \mathcal{L}(\bar{\theta}^{(t)}) \, \Gamma^{(t)})$ **strictly equals zero**. This confirms that our unified bound in Theorem 1 mathematically includes Parallel SGD as a special case and recovers its exact convergence rate **without requiring extra assumptions**.
>
> Following your advice, we have revised **Remark 3** to explicitly discuss this reduction to Parallel SGD, which we provide below.
>
> **Remark 3 (Reduction to Standard Rates)**.
> We consider two special cases where the auxiliary term $A^{(t)}$ vanishes because the consensus error is identically zero ($\Xi\_t \equiv 0$):
> (1) The single-agent case ($m = 1$);
> (2) The fully synchronous Parallel SGD case, where perfect synchronization ensures identical local models ($\theta\_k^{(t)} \equiv \bar{\theta}^{(t)}$).
> In both settings, Theorem 1 naturally recovers the convergence rate of standard (Parallel) SGD, which is of the order $\mathcal{O}({\frac{\sigma^2}{m\varepsilon^2} + \frac{1}{\varepsilon}})$. This confirms that the comparison in Table 1 is fair, as the unified bound applies to these settings without requiring any additional assumptions for the decentralized case.

---

> ### Author Response · Authors · 2025-11-26
> **Response to Reviewer  6EES (Part 2/3)**
>
> > **Q2**: This paper claims that $A_t$ would be negative under Assumption 4 and the condition shown in Proposition 3 holds, but it is still unclear to the reviewer what Assumption 4 implies (Q2.1). Can the authors provide several functions that satisfy Assumption 4 or an intuition to understand Assumption 4 (Q2.2)?
>
> **A2 (Intuition of Assumption 4)**: Thanks for pointing this out. We clarify Assumption 4 by (1) decomposing its mathematical formulation, (2) providing intuition, (3) illustrating with concrete examples, and finally (4) verifying the behavior of decentralized learning on these examples.
>
> **(1) Decomposition:**
>
> To provide a clear intuition, let us first break down the components of the term $S(\theta) \triangleq \text{Tr}(\nabla^2 \mathcal{L}(\theta) \Sigma)$ and the inequality $\nabla \mathcal{L}(\theta)^\top \nabla S(\theta) < 0$.
> - $\nabla^2 \mathcal{L}(\theta)$: The Hessian matrix describing the local curvature of the loss landscape.
> - $S(\theta) = \text{Tr}(\nabla^2 \mathcal{L}(\theta) \Sigma)$: This serves as a generalized measure of "**sharpness**." If $\Sigma = I$ (Identity matrix), $S(\theta) = \sum\_i \lambda_i$, which is the Trace of the Hessian, a standard scalar metric for sharpness. Since $\Sigma$ is a Positive Semi-Definite (PSD) matrix, it acts as a "probe" that weights the curvature along specific directions.
> - $\nabla S(\theta) = \nabla \text{Tr}(\nabla^2 \mathcal{L}(\theta) \Sigma)$: The gradient of the sharpness measure, pointing in **the direction where sharpness increases most rapidly**.
> - **The Inequality**: The condition $\nabla \mathcal{L}(\theta)^\top \nabla S(\theta) < 0$ implies that the inner product of the loss gradient and the sharpness gradient is negative. Geometrically, this means the angle between the direction of loss increase ($\nabla \mathcal{L}$) and sharpness increase ($\nabla S$) are opposing.
>
> **(2) Intuitive Interpretation:**
>
> Let us consider the direction of Gradient Descent (GD), denoted as $v\_{gd} = - \nabla \mathcal{L}(\theta)$. We can rewrite the inequality as:
> $$(-v\_{gd})^\top \nabla S(\theta) < 0 \implies v\_{gd}^\top \nabla S(\theta) > 0$$
> This form makes the intuition immediately clear: As the optimizer moves to **reduce the loss** (along $v_{gd}$), it simultaneously moves in a direction that **increases the sharpness**.
>
> **(3) Illustrating Examples**:
>
> To concretize Assumption 4, we provide specific analytical examples below.
>
> **(3.1) Example A: 1D Trigonometric Function**
>
> Consider the simple function $\mathcal{L}(\theta) = -\cos(\theta)$ over $\theta \in (-\pi, \pi)$.
> - Gradients: The loss gradient is $\nabla \mathcal{L} = \sin(\theta)$. The Hessian is $\nabla^2 \mathcal{L} = \cos(\theta)$.
> - Sharpness Gradient: The sharpness is $S(\theta) = \text{Tr}(\nabla^2 \mathcal{L} \cdot \Sigma) = \sigma \cos(\theta)$. Thus, its gradient is $\nabla S(\theta) = -\sigma \sin(\theta)$.
> - Inner Product:
> $$\nabla \mathcal{L}^\top \nabla S = (\sin(\theta)) \cdot (-\sigma \sin(\theta)) = -\sigma \sin^2(\theta) \leq 0$$
> This confirms that Assumption 4 holds almost everywhere (strictly holds when $\theta \neq k\pi$).
>
> **(3.2) Example B: High-Dimensional Trigonometric Function**
>
> We generalize this to high-dimensional space using the separable function $\theta = (\theta_1, \dots, \theta_d)^\top: \mathcal{L}(\theta) = -\sum\_{i=1}^d \cos(\theta\_i)$:
> - Gradients: The gradient vector is $\nabla \mathcal{L} = [\sin(\theta\_1), \dots, \sin(\theta\_d)]^\top$.
> - Sharpness Gradient: Since the Hessian is diagonal with $H_{ii} = \cos(\theta\_i)$, the sharpness measure is $S(\theta) = \sum\_{i=1}^d \Sigma\_{ii} \cos(\theta\_i)$.
> Taking the gradient w.r.t $\theta$ yields:
> $$\nabla S(\theta) = [-\Sigma\_{11}\sin(\theta\_1), \dots, -\Sigma\_{dd}\sin(\theta\_d)]^\top$$
> - Inner Product:
> $$\nabla \mathcal{L}(\theta)^\top \nabla S(\theta) = \sum\_{i=1}^d \sin(\theta\_i) \cdot (-\Sigma\_{ii}\sin(\theta\_i)) = -\sum_{i=1}^d \Sigma\_{ii} \sin^2(\theta\_i)$$
> This confirms that Assumption 4 strictly holds when $\theta\_i \neq k\pi$ for all $i$, i.e., non-critical points.

---

> ### Author Response · Authors · 2025-11-26
> **Response to Reviewer  6EES (Part 3/3)**
>
> **(4) Verify the Training Dynamics of Decentralized Learning on the Function Provided in (3)**
>
> We verify the training dynamics of Decentralized SGD on the **2D Trigonometric Function** described in Example B (3.2).
>
> **Experimental Setup:**
>
> - Comparison: We compare the trajectory of the globally averaged model trained via Decentralized SGD against a Parallel SGD baseline with 16 agents.
> - Topology and Hyperparameters: A time-varying random topology where each agent communicates with 2 random neighbors per round. Local update steps $H=20$, total communication rounds $R=50$, and learning rate $\eta=0.01$.
> - Noise Model: To mimic stochastic gradient noise, we inject Gaussian noise scaled proportionally to the gradient norm (i.e., gradient-dependent noise).
> Results: The results, averaged over 100 independent random seeds, are presented in the link below.
>
> **Experiment Link:** https://anonymous.4open.science/r/Anonymous-Repo-for-Rebuttal-ICLR26-0E88/Additional%20Experiments/Illustraing_Examples.md
>
> Remarkably, we find that on landscapes satisfying Assumption 4, Decentralized SGD achieves **observable acceleration** compared to the Parallel SGD baseline. This empirical evidence further strengthens our claims and underscores **`a promising future where decentralized training can be leveraged as a superior optimization strategy`**, offering benefits beyond mere communication efficiency.
>
> > **Q3 (Minor Comments)**: The statement of theorems and propositions must explicitly mention which assumptions are used. For instance, Theorem 1 and Propositions 2 and 3 seem to use Assumption 1, but it is not explicitly mentioned in their statements. Additionally, Proposition 2 seems to use Assumption 2, but it is not mentioned in the statement (**Q3.1**). This paper cited the same paper multiple times in the reference. (Kong 2021a and Kong 2021b) (**Q3.2**).
>
> **A3.1**: We thank the reviewer for the careful check regarding the theorem statements.
>
> - Regarding Assumption 1: We clarify that Theorem 1 and Proposition 2 do not rely on Assumption 1, as they do not involve the mixing parameter $p$. Assumption 1 is utilized only in Proposition 3 to derive the sufficient condition.
> - Regarding Assumption 2: We agree with the reviewer. We have carefully revised the statement of Proposition 2 in the manuscript to explicitly include Assumption 2, ensuring mathematical rigor and clarity.
>
> **A3.2 (Citations)**: Thanks and addressed.

---

### Official Review · Reviewer_EC7X · 2025-10-30

**Soundness:** 3
**Presentation:** 3
**Contribution:** 3
**Rating:** 8
**Confidence:** 3

**Summary:**

This paper looks at how to make communication more efficient in decentralized distributed training, where there is no central node and the network connections are random. The authors study when communication should happen during training, checking if communicating more at certain times helps the final shared model perform better.

In their experiments, they allow heavy communication only during a specific part of training, while at other times, each node talks to just one random neighbor. The results show that increasing communication near the end of training works best.

The paper also includes a theoretical analysis that proves their method converges. It further shows that training can be faster when certain gradient patterns are present.

**Strengths:**

The paper makes an interesting analysis of mergability of models trained on different (heterogeneous) subsets of data. Theoretical analysis shows that it depends on the sharpness of the merging point, which coincides with the usual intuition of how loss surface behaves. Moreover, proposed approach helps to reduce not needed communication to minimum.

**Weaknesses:**

Investigation of the communication after convergence is missing - in particular Figure2 demonstrates that continued individual training after full merge leads to reduction of the performance, the question is does it ever stop? Can the model converge to a state that neither merging, no continuing training changes the resulting performance?

**Questions:**

1 - Can you align your analysis with the dynamic averaging approach in [1], which claims that communication after convergence is not needed, motivating it also through the properties of loss surface; in particular, when models are already in one convex hull their performance will be basically same and aggregation is not needed anymore (I understand that there the setup is federated learning, but I would guess that this is the general understanding of how individually trained models behave).

[1] Kamp, Michael, et al. "Efficient decentralized deep learning by dynamic model averaging." Joint European conference on machine learning and knowledge discovery in databases. Cham: Springer International Publishing, 2018.

---

> ### Author Response · Authors · 2025-11-26
> **Response to Reviewer EC7X (Part 1/2)**
>
> We sincerely thank the reviewer for their strong support and for recognizing the value of our theoretical analysis and the communication scheduling approach. We address your remaining questions below.
>
> > **Q1 (Comparison and Geometric Intuition)**: Can you align your analysis with the dynamic averaging approach in [1], which claims that communication after convergence is not needed, motivating it also through the properties of loss surface; in particular, when models are already in one convex hull their performance will be basically same and aggregation is not needed anymore (I understand that there the setup is federated learning, but I would guess that this is the general understanding of how individually trained models behave).
>
> **A1 (Comparison and Geometric Intuition)**: We thank the reviewer for highlighting reference [1] and for this insightful question.
>
> We agree that in the IID setting considered in [1], local models tend to converge to the same low-loss basin. In such cases, once models enter this shared region (aligning with the "convex hull" intuition), they are functionally similar, and further aggregation indeed brings diminishing returns.
>
> However, the dynamics are fundamentally different in the **heterogeneous (Non-IID) settings** we consider. In Non-IID scenarios, the local loss landscapes differ significantly across agents. As illustrated in the following Figure, training becomes a constant **"tug-of-war"** between two forces:
>
> - **Local Optimization Force**: Driven by distinct local distributions, this force pulls each agent towards its own local basin, causing models to drift apart.
> - **Global Aggregation Force**: Provided by communication, this force pulls agents back towards a consensus.
>
> **Figure Link (Geometric Intuition):** https://anonymous.4open.science/r/Anonymous-Repo-for-Rebuttal-ICLR26-0E88/Local%20and%20Global%20Forces%20in%20Decentralized%20Learning%20Loss%20Landscape.md
>
> Consequently, distinct from the IID case, local models do not necessarily reach consensus. Instead of settling into a single convex hull, local models remain dispersed on the *"high-loss ring"* shown in Figure 1(c) in our paper where they are constantly pulled apart by heterogeneous local gradients.
>
> We thank the reviewer for suggesting this geometric perspective. This analysis leads to two key implications:
>
> - **Communication is a Counter-Force**: Communication remains necessary even near global convergence, **not merely to reduce variance** (as in IID cases), but to *counteract* local optimization forces and *“anchor”* the global model within the optimal global basin.
> - **Effectiveness of Single Merging**: This strengthens our finding that single global merging works effectively even when local models do not reach consensus.  As we discussed in Section 4.2, we identify an emergent geometric structure where decentralized training can surprisingly *“guides”* local models to a *ring-like high-loss region* surrounding a *central low-loss basin*.
>
> **`These observations point to a promising future for distributed training.`** Instead of enforcing strict consensus via expensive communication, we can **save communication and allow local models to maintain a "loosely coupled" state**, where they explore locally while *"anchored"* to the global basin. A simple final merge can then effectively *“recover”* the performance of communication-intensive training.
>
> ### Reference
>
> [1] Kamp, Michael, et al. "Efficient decentralized deep learning by dynamic model averaging." Joint European conference on machine learning and knowledge discovery in databases. Cham: Springer International Publishing, 2018.

---

> ### Author Response · Authors · 2025-11-26
> **Response to Reviewer EC7X (Part 2/2)**
>
> > **Q2**: Investigation of the communication after convergence is missing - in particular Figure 2 demonstrates that continued individual training after full merge leads to reduction of the performance, the question is does it ever stop (**Q2.1**)? Can the model converge to a state that neither merging, no continuing training changes the resulting performance (**Q2.2**)?
>
> **A2.1**: Thanks for the insightful suggestion. This does not stop, a consistent reduction of performance is observed of continued individual training after full merge. As suggested, we further conducted additional experiments using ResNet-18 on the Tiny ImageNet dataset, **doubling the number of training rounds**.
>
> **Results**: The observations remain consistent with our main paper: the global performance drops after switching full merge back to normal decentralized training with limited random communication. The results are provided in this **[link](https://anonymous.4open.science/r/Anonymous-Repo-for-Rebuttal-ICLR26-0E88/Additional%20Experiments/Merge2Local.md)**
>
> **Experiment Link:** https://anonymous.4open.science/r/Anonymous-Repo-for-Rebuttal-ICLR26-0E88/Additional%20Experiments/Merge2Local.md
>
> **(Explanation)** We explain the mechanism of this phenomenon as follows:
>
> Even when the global merge successfully places the model in the central "global basin", this location is not a stationary point for the individual agents due to data heterogeneity. We analyze the mechanism of performance drop below:
> - **Global Merging**: The global merge strengthens the global aggregation force, *“pulling”* dispersed models from the *“high-loss ring"* into the global basin.
> - **Continued Training**: If we subsequently resume local training, the local optimization force immediately dominates the update direction. This force pulls the local models out of the global basin towards the dispersed "high-loss ring."
>
> **A2.2 (Rethinking "Convergence" in Decentralized Learning)**: Building on the "tug-of-war" intuition established in **A1**, such a state does not exist under heterogeneous settings without full communication.
> Reviewer's hypothesis implies a state where the model is simultaneously stable for local optimization and global aggregation. We clarify the inherent conflict between these conditions in heterogeneous settings:
> - **If models are in the "balanced state"**: In this state, local models remain dispersed on the "high-loss ring", where they preserve high *“mergeability”*. Here, merging changes performance significantly by strengthening the global aggregation force to pull them into the global basin.
> - **If models are in the global basin:** In this case, continuing training changes performance (degrades it). Since this state is not a stationary point for individual agents, the local optimization force dominates and pulls the model out of the global basin towards the dispersed "high-loss ring."
>
> Therefore, a static equilibrium satisfying both conditions remains elusive due to inherent data heterogeneity. The system is essentially **dynamic**, governed by the conflict between local optimization and global aggregation.
> Crucially, our observation highlights a broader implication of our work: it suggests the need to **rethink the definition of "convergence" in fully decentralized learning, moving beyond strict static consensus to recognize such 'dynamic stability' as a valid and effective state**.

---

### Author Response · Authors · 2025-12-03
**Summary of Rebuttal (Part 1/3)**

Dear ACs,

We sincerely appreciate your time and effort in handling our submission. For your convenience, we provide a concise summary of our rebuttal below. We also thank the reviewers for their constructive feedback, which has significantly strengthened our paper. Our rebuttal highlights the following points:

* **Reviewer EC7X** suggested aligning our analysis with Kamp et al. [1] regarding the necessity of communication after convergence and inquired about the system's dynamics if training continues after merging. We addressed this by distinguishing our heterogeneous setting via the "tug-of-war" mechanism and providing experiments showing that decentralized learning exhibits a novel dynamic stability rather than a static state.
* **Reviewer 6EES** questioned the fairness of comparing our convergence rates with Parallel SGD and the intuition for the "Progressive Sharpening" assumption. We responded by demonstrating that the comparison is **strictly fair**, showing that our bounds exactly recover standard Parallel SGD rates in the Parallel SGD setting (added in Remark 3). We validated the sharpening assumption using high-dimensional trigonometric functions as analytical examples.
* **Reviewers pdQg & jbK5** focused on the practicality of the global merging operation and the feasibility of **topology-constrained merging**. We addressed this by validating a practical implementation where global merging is replaced by iterative gossip averaging, demonstrating that **just 5 rounds of gossip** are sufficient to maintain the significant performance gains.

We also provide a more detailed version of our responses below:

---

> ### Author Response · Authors · 2025-12-03
> **Summary of Rebuttal (Part 2/3)**
>
> ## **`Reviewer EC7X`**
>
> **Q1 (Comparison + Geometric Intuition)**: The reviewer asked to align our analysis with [1], which claims that communication is unnecessary after convergence because models settle in a shared convex hull.
>
> **Our Response**: We explained that unlike the IID setting in [1], heterogeneous environments create a **["tug-of-war" dynamics (link)](https://anonymous.4open.science/r/Anonymous-Repo-for-Rebuttal-ICLR26-0E88/Local%20and%20Global%20Forces%20in%20Decentralized%20Learning%20Loss%20Landscape.md)** between local optimization and global aggregation forces, keeping models on a *"high-loss ring"* while preserving their *“mergeability”*. **`This geometric insight points to a promising future for distributed training`**, where we can save communication costs while maintaining performance by allowing "loosely coupled" models to explore locally while being effectively *“anchored”* to the global basin.
>
> **Q2 (Post-Merge Dynamics)**: The reviewer suggested that an investigation of communication after convergence was missing and asked if the performance drop stops or reaches a stable state if training continues after merging.
>
> **Our Response**: We provided [additional experiments showing the performance drop persists (link)](https://anonymous.4open.science/r/Anonymous-Repo-for-Rebuttal-ICLR26-0E88/Additional%20Experiments/Merge2Local.md). We explained this phenomenon using the **"tug-of-war"** intuition established in **Q1**: resuming local training allows local optimization forces to pull the model out of the global basin back towards the “dispersed ring-like regions”. We clarified that in heterogeneous settings, no static state satisfies both local and global objectives; rather, the system exhibits a novel **dynamic stability**, suggesting a need to **rethink the definition of "convergence"** in decentralized learning.
>
> ## **`Reviewer 6EES`**
>
>
> **Q1 (Fairness of Comparison with Parallel SGD)**: The reviewer argued that our claim regarding Decentralized SGD matching Parallel SGD rates is an overclaim, suggesting the comparison is unfair because Theorem 1 depends on $\Xi\_t$.
>
> **Our Response**: We clarified that our comparison is **fair and holds under the same assumptions**. In the case of Parallel SGD, the consensus error terms $\Gamma^{(t)}$ and $\Xi\_t$ are identically zero by definition due to perfect synchronization. Consequently, the complex auxiliary term $A\_t$ in Theorem 1 vanishes, exactly recovering the standard Parallel SGD rate $\mathcal{O}({\frac{\sigma^2}{m\varepsilon^2} + \frac{1}{\varepsilon}})$ without extra assumptions. We have added **Remark 3** to explicitly formalize this reduction.
>
> **Q2 (Intuition of Assumption 4)**: The reviewer asked for the intuition behind Assumption 4 (Progressive Sharpening) and requested examples of functions that satisfy it.
>
> **Our Response**: We explained that Assumption 4 (i.e., $\nabla \mathcal{L}^\top \nabla Tr{(H \Xi)} < 0$) intuitively means that as the optimizer reduces loss, it moves into sharper regions of the landscape, a phenomenon widely observed  in deep learning. We provided analytical examples, such as **high-dimensional trigonometric functions**, where this condition strictly holds. **Furthermore, we verified our theory by running decentralized SGD on these functions**, observing a surprising **[empirical acceleration over Parallel SGD (link)](https://anonymous.4open.science/r/Anonymous-Repo-for-Rebuttal-ICLR26-0E88/Additional%20Experiments/Illustraing_Examples.md)**, which points to **`a promising future where decentralized training may serve as a superior optimization strategy`**.
>
> **Q3 (Minor Comments)**: The reviewer requested clarification on which assumptions are used in the theorem statements.
>
> **Our Response**: Addressed. We clarified that Assumption 1 is only needed for Proposition 3, while we have updated Proposition 2 to explicitly state its reliance on Assumption 2.

---

> ### Author Response · Authors · 2025-12-03
> **Summary of Rebuttal (Part 3/3)**
>
> ## **`Reviewer pdQg`**
>
> **Q1 (Topology-Constrained Merging)**: The reviewer asked whether our main conclusions hold under realistic network topologies where only random communication or topology-constrained merging is possible, rather than a single global merge.
>
> **Our Response**: We confirmed that **our findings are robust** by conducting additional experiments where agents communicate via a general topology**. Results show that **topology-constrained merging** (e.g., just 5 rounds of gossip) effectively bridges the gap to ideal global merging, maintaining the **significant performance improvement** detailed in this **[experiment link](https://anonymous.4open.science/r/Anonymous-Repo-for-Rebuttal-ICLR26-0E88/Additional%20Experiments/Final_Gossip_Merging.md)**.
>
> **Q2 (Intuition of Proposition 3)**: The reviewer asked for intuition regarding Proposition 3, specifically why communication is needed when the gradient norm is small.
>
> **Our Response**: We clarified that the proposition refers to the **global gradient** (which can be small) versus **local gradients** (which remain large due to data heterogeneity), creating a conflict which captures the **'tug-of-war' intuition**. We provided a **[visual illustration](https://anonymous.4open.science/r/Anonymous-Repo-for-Rebuttal-ICLR26-0E88/Local%20and%20Global%20Forces%20in%20Decentralized%20Learning%20Loss%20Landscape.md)** showing that stronger communication is crucial at the final stage to counteract local drift and maintain the model in the global basin.
>
> ## **`Reviewer jbK5`**
>
> **Q1 (Topology-Constrained Merging)**: The reviewer suggested replacing the "all-reduce" global merge with iterative gossip averaging.
>
> **Our Response**: Consistent with our response to **Reviewer EC7X**, we confirmed via new experiments that replacing global merge with just 5 rounds of gossip averaging maintains the performance gains, as detailed in this **[experiment link](https://anonymous.4open.science/r/Anonymous-Repo-for-Rebuttal-ICLR26-0E88/Additional%20Experiments/Final_Gossip_Merging.md)**.
>
> **Q2 (Centralized Baseline)**: The reviewer requested the addition of a centralized SGD training curve to Figure 3c to serve as a baseline.
>
> **Our Response**: We have updated Figure 3c to include the centralized SGD baseline as requested. The revised comparison demonstrates that our decentralized method achieves convergence speed comparable to the centralized baseline, available in this **[experiment link](https://anonymous.4open.science/r/Anonymous-Repo-for-Rebuttal-ICLR26-0E88/Additional%20Experiments/Centralized_Baseline.md)**.
>
> **Q3 (Comparison with Aketi et al. [2])**: The reviewer suggested that our findings on low communication preserving mergeability were previously shown in Aketi et al. [2] (Sparse-Push).
>
> **Our Response**: We clarified that while [2] focuses on **gradient sparsification** with single local steps ($H=1$), our work investigates **topological sparsification** under the much more challenging regime of many local steps ($H=100$). We highlighted that our contribution lies in discovering that mergeability persists even under this high heterogeneity, explained by our unique geometric analysis of the "high-loss ring."
>
> ### Reference
>
> [1] Kamp, Michael, Boley, Mario, Missura, Olana, & Gärtner, Thomas. "Efficient decentralized deep learning by dynamic model averaging." Joint European conference on machine learning and knowledge discovery in databases. Cham: Springer International Publishing, 2018.
>
> [2] Aketi, Sai Aparna, Singh, Abhiroop, & Rabaey, Jan. "Sparse-push: Communication-& energy-efficient decentralized distributed learning over directed & time-varying graphs with non-iid datasets." arXiv preprint arXiv:2102.05715, 2021.

---

### Meta-Review · Area_Chair_mkj5 · 2025-12-30

**Summary:**

The paper starts with the empirical observation that a single global merging of decentralized models, even under severely constrained communication and high data heterogeneity, can significantly improve global generalization.
On the theoretical side, they prove that the globally merged model of decentralized SGD can match the rate of parallel SGD. They also explain that limited communication is sufficient to ensure mergeability and that intensive communication in the later stages of training is sufficient for effective learning.

**Reviewer Concerns:**

Overall, the reviewers were very positive about the quality of the writing, the interest of the empirical observations, and the theoretical results. Very few criticisms were raised, but the authors responded precisely and took the remarks very seriously.

**Reviewer Scores:**

The scores were already high enough before the rebuttal, so I don’t think they would have changed. Perhaps pdQg could have gone from a 6 to an 8.

---

### Decision · Program_Chairs · 2026-01-26

Accept (Oral)